# Self-inactivating, all-in-one AAV vectors for precision Cas9 genome editing via homology-directed repair in vivo

Raed Ibraheim [1,9], Phillip W. L. Tai [2,3], Aamir Mir[1,9], Nida Javeed[1], Jiaming Wang[2], Tomás C. Rodríguez[1], Suk Namkung[2], Samantha Nelson[3], Eraj Shafiq Khokhar[1], Esther Mintzer[4], Stacy Maitland[4], Zexiang Chen[1], Yueying Cao[1], Emmanouela Tsagkaraki [5,6], Scot A. Wolfe [4,7], Dan Wang [1,2], Athma A. Pai[1], Wen Xue [1,4,5,7], Guangping Gao [2,3,7,8] & Erik J. Sontheimer [1,5,7 ✉]

Adeno-associated virus (AAV) vectors are important delivery platforms for therapeutic genome editing but are severely constrained by cargo limits. Simultaneous delivery of multiple vectors can limit dose and efficacy and increase safety risks. Here, we describe single-vector, ~4.8-kb AAV platforms that express Nme2Cas9 and either two sgRNAs for segmental deletions, or a single sgRNA with a homology-directed repair (HDR) template. We also use anti-CRISPR proteins to enable production of vectors that self-inactivate via Nme2Cas9 cleavage. We further introduce a nanopore-based sequencing platform that is designed to profile rAAV genomes and serves as a quality control measure for vector homogeneity. We demonstrate that these platforms can effectively treat two disease models [type I hereditary tyrosinemia (HT-I) and mucopolysaccharidosis type I (MPS-I)] in mice by HDR-based correction of the disease allele. These results will enable the engineering of single-vector AAVs that can achieve diverse therapeutic genome editing outcomes.

[1] RNA Therapeutics Institute, University of Massachusetts Medical School, Worcester, MA 01605, USA. [2] Horae Gene Therapy Center, University of Massachusetts Medical School, Worcester, MA 01605, USA. [3] Department of Microbiology and Physiological Systems, University of Massachusetts Medical School, Worcester, MA 01605, USA. [4] Department of Molecular, Cell and Cancer Biology, University of Massachusetts Medical School, Worcester, MA 01605, USA. [5] Program in Molecular Medicine, University of Massachusetts Medical School, Worcester, MA 01605, USA. [6] University of Crete School of Medicine, Heraklion, Crete 71003, Greece. [7] Li Weibo Institute for Rare Diseases Research, University of Massachusetts Medical School, Worcester, MA 01605, USA. [8] Viral Vector Core, University of Massachusetts Medical, School, Worcester, MA 01605, USA. [9] Present address: Tessera Therapeutics, Inc., 200 Sidney Street, Cambridge, MA 02139, USA. ✉email: erik.sontheimer@umassmed.edu

Bacterial and archaeal adaptive immune systems are known as clustered, regularly interspaced, short, palindromic repeats (CRISPR), along with their CRISPR-associated (Cas) proteins, have been at the center of the RNA-guided genome engineering revolution[1,2], including genome editing[3–11]. The most widely adopted CRISPR-Cas genome editing platforms[12,13] are the Cas9 endonucleases that create double-strand breaks (DSBs) in target DNA sequences complementary to a designed single-guide RNA (sgRNA). In addition to the sgRNA, Cas9 requires the presence of a protospacer adjacent motif (PAM), which is a specific sequence proximal to the sgRNA-complementary target sequence[1,2,14,15]. Once cleaved, the cell relies on the DNA repair machinery to resolve these DSBs through non-homologous end joining (NHEJ), microhomology-mediated end joining (MMEJ), or homology-directed repair (HDR), the latter of which can enable a wide range of precisely determined repair outcomes[16]. More recently, base editing[17] and prime editing[18] have emerged as promising technologies for achieving DSB-independent precision genome editing[13].

The most widely used Cas9 homolog has been *Streptococcus pyogenes* Cas9 [SpyCas9, 1368 amino acids (aa)][1,19], which exhibits robust editing efficiencies, as well as a broad targeting range due to its short 5′-NGG-3′ PAM[1]. Additional targeting events have been enabled by SpyCas9 engineering to alter or reduce PAM requirements, and by identifying additional Cas9 homologs that are also active in eukaryotic cells but require distinct PAMs[13]. Some such Cas9 homologs, including Sth1Cas9 (1120 aa)[3], Nme1Cas9 (1082 aa)[20,21], SauCas9 (1053 aa)[22], Cje-Cas9 (984 aa)[23], Nme2Cas9 (1082 aa)[24], and SauriCas9 (1061 aa)[25] are substantially smaller. The reduced sizes are important for some modes of in vivo delivery, especially viral vectors such as AAV, which has great potential as a delivery modality but is severely constrained in cargo size (<5 kb in most cases)[26]. Delivery of most of these compact Cas9s within single-vector AAVs, in which both the Cas9 and its sgRNA are packaged into the same virion, has been validated in vivo[22–24,27–29]. AAV vectors encoding SauCas9 along with two sgRNAs[30,31], as well as CjeCas9 with three sgRNAs[32], have also been reported, enabling the introduction of multiplexed edits or segmental deletions. A few of the smaller Cas9s are also less prone to off-target cellular editing than wild-type SpyCas9[23,24,27,33,34], even with guides that support efficient on-target editing, likely because of reduced enzymatic efficiencies that disproportionately dampen activity at near-cognate sites[35].

Despite the positive attributes conferred by Sth1Cas9, Nme1-Cas9, SauCas9, and CjeCas9, they suffer from a more restricted targeting range than what is imparted by SpyCas9, due to their more complex PAMs that are less abundant throughout the genome. Engineering of the PAM-interacting domain of SauCas9 reduced its wild-type PAM requirements (5′-NNGRRT-3′)[22] to 5′-NNNRRT[36], helping to expand its targeting flexibility. More recently, Nme2Cas9 (5′-NNNNCC-3′ PAM)[24], SauriCas9 (5′-NNGG-3′ PAM)[25], and SlugCas9 (5′-NNGG-3′ PAM)[37] were discovered as natural compact Cas9s with dinucleotide PAMs and robust editing activity in mammalian cells. To our knowledge, the in vivo utility of SauriCas9 and SlugCas9 with all-in-one AAV vectors is yet to be documented.

To date, in vivo genome editing applications by recombinant AAV (rAAV)-encoded Cas9 have mostly been limited to gene knockouts or insertions by cleavage events that are repaired by NHEJ pathways. Precise editing that introduces specific sequence changes that are not compatible with microhomology-mediated end joining (MMEJ) repair[38,39] are very difficult to attain via HDR in vivo due in part to the requirement for co-delivery of a repair template, in addition to the low or absent activity of the HDR pathway in non-cycling cells, or during the G1 phase of the

cell cycle[16]. Co-delivery of multiple AAV vectors for HDR-based precise editing in vivo, e.g., with the Cas9 effector on one vector, the donor on another, and the sgRNA(s) encoded by either or both vectors, has shown some promise in animal models[40–42]. Other strategies for precise editing in vivo, such as base editing and prime editing, usually involve Cas9-fused effectors that are sufficiently large as to again require two vectors[43–47]; although single-AAV adenine base editing systems are beginning to emerge[48], to our knowledge they have not yet been validated in vivo, and many disease alleles (including indels and transversions) are not amenable to correction by A:T-to-G:C editing. Importantly, the combined AAV dosages required for multi-vector strategies raise significant concerns about safety during clinical translation[49,50], especially in light of recent events in human trials of a high-dose AAV gene therapy[51]. A recent report described a single-AAV system in which SauCas9, two sgRNAs, and an HDR donor were designed into a single >5 kb vector construct[52]. However, this study used an AAV8 capsid that lacks the VP2 subunit, yielding low vector titers with the increased cargo size. Precise genome editing strategies compatible with single-vector AAVs using well-established capsids that can be produced at preclinical and clinical scales are therefore still needed.

In this work, we build upon the compact size, high cellular editing accuracy, and targeting flexibility of Nme2Cas9[24,53]. We have systematically engineered all-in-one AAV:Nme2Cas9 vector systems with increased utility beyond that afforded by our initial version[24], which encoded the effector and just one sgRNA. First, we showed that minimizing the Nme2Cas9 cassette, its regulatory elements, and the sgRNA provided additional coding capacity that could be used to incorporate a second sgRNA. Specific arrangements and orientations of the dual-sgRNA system were validated for their capacity to be packaged into a single-AAV vector. To achieve this, we developed a nanopore-based sequencing approach to directly analyze the packaged vector genome to ensure the homogeneity of rAAV vectors. As a proof-of-principle demonstration, we used these dual-sgRNA vectors in vivo to introduce two DSB and induce exon excision or inversion to rescue the disease phenotype of type I hereditary tyrosinemia (HT-I) mice. We then demonstrated that one of the two sgRNA cassettes could be replaced by an HDR donor while remaining below the 5-kb cargo size required for efficient packaging into standard serotypes. We further endowed this system with Nme2Cas9 target sites flanking the HDR donor to allow self-inactivation of the vector as a function of editing effector accumulation. To enable vector packaging of a self-targeting nuclease system, we employed an anti-CRISPR (Acr) protein[54,55] to prevent self-cleavage during vector cloning and packaging. The in vivo efficiency of this all-in-one HDR-based Nme2Cas9 editing system was validated in two mouse models of inherited disease [the metabolic disorder HT-I and the lysosomal storage disease mucopolysaccharidosis type I (MPS-I)]. In both cases, precise editing efficiencies reached levels sufficient for therapeutic benefit. The establishment of a self-inactivating single-rAAV system capable of precise, versatile, HDR-based genome sequence changes in vivo, combined with the accuracy and wide targeting range of Nme2Cas9, expand our capabilities to develop more potent therapeutic genome editing platforms.

## Results

**Development of a gene-editing rAAV construct with a minimized Nme2Cas9 payload.** We previously engineered ~4.7-kb rAAV gene editing vectors that deliver Nme1Cas9 or Nme2Cas9 and a 145-nucleotide (nt) sgRNA[24,27]. Both Cas9 transgenes were fused with four nuclear localization signals (NLSs) and a triple-

HA epitope tag. To engineer a single Nme2Cas9 vector with additional capabilities, we created a shortened Nme2Cas9 + sgRNA backbone. Directed by Nme2Cas9 structural information[56] as well as empirical testing of guide requirements, we truncated the repeat:anti-repeat duplex to create two shortened 121-nt and 100-nt sgRNAs (Nme.sgRNA-121 and Nme.sgRNA-100, respectively) (Supplementary Fig. 1). The truncated guides were cloned into the previously published Nme2Cas9 AAV backbone[24]. Comparison of editing efficacy at the *CYBB*, *LOC100505797*, and *Fah* genomic loci by plasmid transfection into HEK293T (human) and Neuro2a (mouse) cells showed that Nme.sgRNA-100 partially compromises Nme2Cas9 activity, while Nme.sgRNA-121 performance is more consistent and comparable to that of full-length Nme.sgRNA (Supplementary Fig. 1). Although Nme.sgRNA-100 performance was less consistent by plasmid transfection when driven by a U6 promoter, ex vivo editing by Nme2Cas9 following ribonucleoprotein (RNP) delivery using a chemically synthesized Nme.sgRNA-100 was robust. We systematically compared editing efficiencies using full-length and 100-nt sgRNAs that were made by T7 in vitro transcription (IVT), with and without calf intestinal alkaline phosphatase (CIP) treatment[57], and commercial synthetic guides carrying three 2′-*O*-methyl phosphorothioate modifications at each end (synthesized by Synthego, USA) to protect against exonucleases[58]. Our editing data in HEK293T cells and the difficult-to-transfect T lymphoblast cell line MOLT-3 showed that the truncated Nme.sgRNA-100 conferred a higher editing efficiency compared to the other sgRNA formats when delivered as electroporated RNPs (Supplementary Fig. 1). Shortening to 100 nt places Nme2Cas9 sgRNAs within the size range needed for facile chemical synthesis for RNP delivery.

Because the 121-nt sgRNA version was more consistent as an intracellular transcript, we used it to create a shortened AAV:Nme2Cas9 vector. Using Nme.sgRNA-121 (henceforth referred to as sgRNA), a short polyadenylation signal[59], and a Nme2Cas9 open reading frame (ORF) with two NLSs and no epitope tags, we generated a minimized AAV:Nme2Cas9:sgRNA vector with a total genome size of ~4.4 kb (Supplementary Fig. 1; Supplementary Note). We used plasmid transfection into HEK293T and Neuro2a cells to compare the editing activity of the shortened construct to that of the previously published 4.7-kb AAV:Nme2Cas9:sgRNA backbone. Sequencing analysis indicated that the shortened AAV backbone is functional in cells with comparable editing efficiencies at three of the four sites tested (Supplementary Fig. 1). Overall, our results confirmed that the shortened AAV:Nme2Cas9 construct is efficient at genome editing, affording additional cargo capacity to expand functionality.

**Design of dual-sgRNA AAV:Nme2Cas9 vectors**. Previous reports have shown that when two simultaneous DSBs occur, the NHEJ pathway can religate the free DNA ends to produce a segmental deletion[31,60–63]. The delivery of two guides within the same AAV vector can support this form of editing, facilitate the simultaneous knockout of two distinct targets, or enable the use of paired nickases via single-rAAV delivery[30]. The delivery of two sgRNAs could be especially important as a potential therapeutic route to address microsatellite expansion diseases, such as Friedreich's ataxia, Huntington's disease, and Fragile X syndrome, by excising this mutated gene fragment. However, as with strong secondary structural elements in general[64,65], the inclusion of two sgRNA cassettes in the same backbone can present significant obstacles to such designs by inducing the formation of truncated vector genomes[66]. To address such events, we engineered AAV:Nme2Cas9 constructs carrying two sgRNA-121 cassettes in

various positions and orientations (Fig. 1a). Each of these four backbones was packaged into an AAV8 capsid and the viral DNA was isolated[67]. Size comparisons using alkaline agarose gel electrophoresis analysis revealed that depending on the sgRNA-121 cassette orientation and position, the genomes of certain dual-sgRNA AAV vectors indeed produced multiple truncated forms. Two designs (henceforth referred to as Dual-sgRNA:Designs 1 and 4; Supplementary Note) appear to contain predominantly full-length, non-truncated ~4.8-Kb genomes (Fig. 1b).

To provide a more comprehensive profile of AAV genome homogeneity, we developed a real-time, nanopore-based sequencing platform to survey individual genomes of these dual-sgRNA AAV:Nme2Cas9 vectors. Our sequencing data showed that unlike the Dual-sgRNA:Design 4 vector, which predominantly contains full-length genomes from ITR to ITR, Dual-sgRNA:Design 1 yields a high frequency of truncated genomes (Fig. 1c, d). These truncations are predominantly centered at two positions. The first is positioned at the sgRNA expression cassette, resulting in vectors that only carry a single sgRNA. The second is centered at the 3′ end of the hNme2Cas9 transgene.

To test the efficacy of these constructs for the induction of segmental deletion, we tested Design 1 and Design 4 for their capacity to excise a 606-bp fragment spanning exons 3 and 4 of the *Hpd* gene (Fig. 1e). We did this first in cultured cells and then in an HT-I mouse model in which the *Fah* gene, which encodes the liver enzyme fumarylacetoacetate hydrolase (FAH), is disrupted by the insertion of a neomycin cassette in exon 5 (*Fah^neo/neo*)[68]. A hallmark of this disease in mice is the gradual loss of body weight when the mice are not maintained on water supplemented with the drug 2-(2-nitro-4-trifluoromethylbenzoyl)-1,3-cyclohexanedione (NTBC), which blocks the tyrosine catabolism pathway upstream of FAH, preventing the buildup of the hepatotoxic metabolites that result from loss of FAH[69]. Knocking out HPD by excising exons 3 and 4 can rescue the lethal phenotypes of HT-I mice by blocking the tyrosine catabolic pathway upstream of FAH[27,70]. In addition to the expected segmental DNA deletion, we also anticipated detecting inversions and small indels at one or both of the two target sites[31,63,71]. Initially we transfected Dual-sgRNA:Design 1 and Dual-sgRNA:-Design 4 plasmids into mouse Neuro2a cells and used single molecule, real-time (SMRT) sequencing to assess editing efficiency, using primers that included unique molecular identifiers (UMIs) to enable correction for amplification bias with products of different lengths. Our analysis showed that both of these plasmids efficiently yielded the expected segmental deletion only when spacers gRNA-I and gRNA-II are co-expressed. Our data also revealed that localized, small insertions/deletions (indels) are generated at both target sites when segmental deletion does not occur (Supplementary Figs. 1 and 2). In addition to confirming that co-expression of Nme2Cas9 and two sgRNAs can efficiently induce segmental deletions in cultured cells[24], these results also affirm that dual-sgRNA AAV backbones should be carefully evaluated to avoid the formation of truncated genomes that may compromise vector performance in vivo[66].

**Validation of dual-sgRNA AAV:Nme2Cas9 gene editing in vivo**. To test the ability of dual-sgRNA AAV:Nme2Cas9 vectors to disrupt *Hpd* in HT-I mice, we used the two validated AAV8:Nme2Cas9 Dual-sgRNA:Design 1 and 4 vectors (Fig. 1c, d) to induce segmental deletion of exons 3 and 4 in vivo in *Fah^neo/neo* mice. Adults were injected with $4 \times 10^{11}$ gc of each vector via tail vein. To assess baseline editing efficiency without selective clonal expansion of edited hepatocytes, we evaluated editing in one mouse from each group prior to NTBC withdrawal, as well as in C57BL/6 mice (Fig. 2a). After NTBC withdrawal,

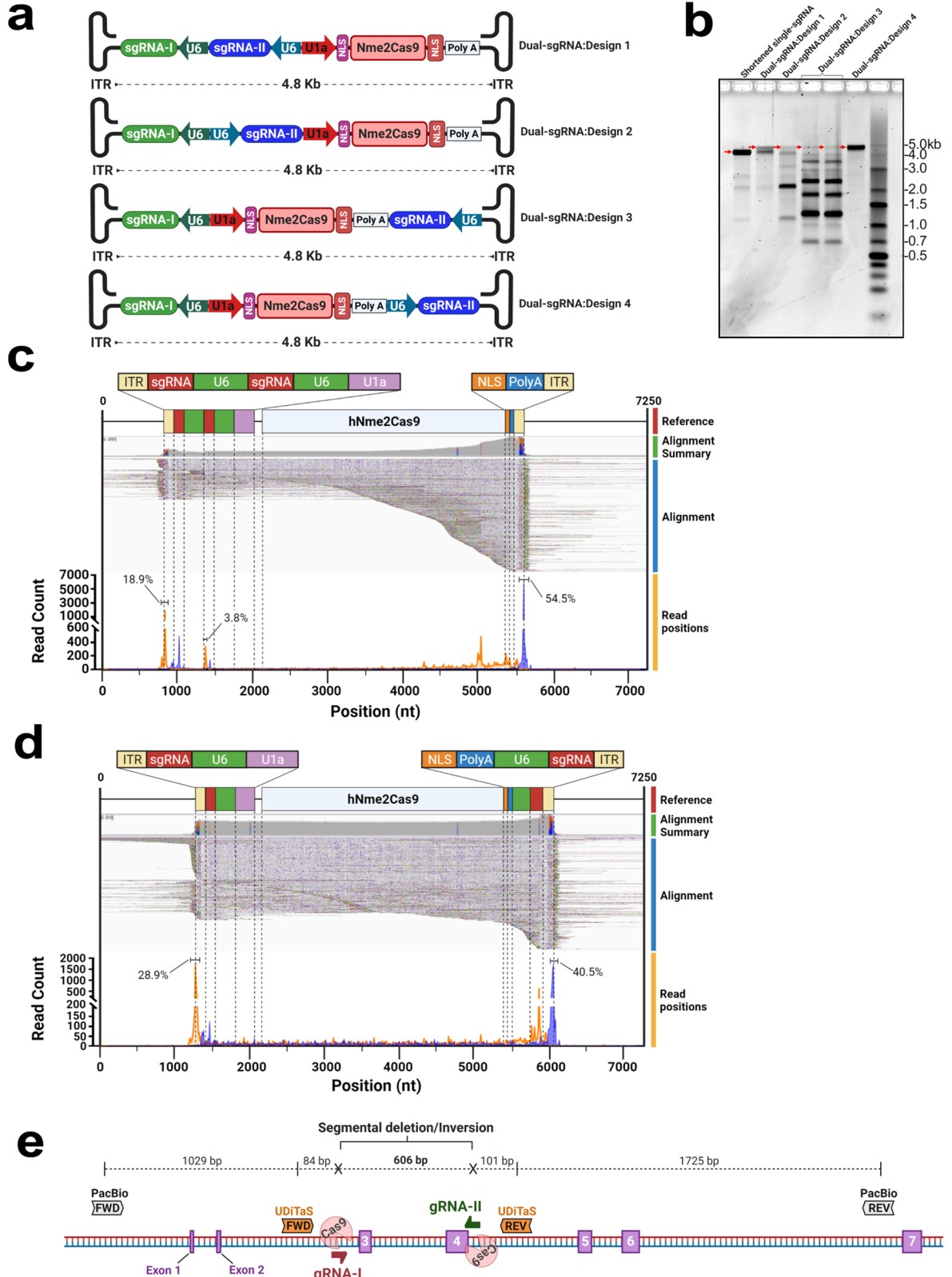

*Fah^neo/neo* mice were effectively rescued of the lethal phenotypes of HT-I as indicated by body weight stability for 3 weeks (Fig. 2b). In contrast, PBS-injected *Fah^neo/neo* mice lost over 10% of their body weights within 4–13 days of NTBC withdrawal and were euthanized (Fig. 2b).

After completion of editing regimens, we harvested liver DNA and performed UMI-augmented SMRT sequencing of target-spanning amplicons for all mice in the study to assess the efficiencies of segmental deletion and other outcomes. These analyses revealed that segmental deletions were successfully induced by Nme2Cas9 in vivo with both dual-sgRNA vector designs (Fig. 2c), with the expected 3.0-kb read lengths resulting from deletion of the 606-bp fragment from the 3.6-kb amplicon (Supplementary Fig. 2). In C57BL/6 mice, which are not expected

**Fig. 1 In vitro design and Nanopore sequencing validation of dual-sgRNA AAV:Nme2Cas9 vectors. a** Schematic of four different designs of dual-sgRNA AAV:Nme2Cas9 vector plasmids containing two sgRNA expression cassettes. **b** Alkaline agarose gel electrophoresis of viral DNA isolated from the four dual-sgRNA AAV:Nme2Cas9 vectors presented in (**a**) after packaging in AAV8. DNA size markers are indicated. Red arrows indicate full-length non-truncated viral genomes. This analysis was performed once. **c, d** Alignments of Nanopore sequencing reads representing DNAs extracted from Dual-sgRNA:Design 1 (**c**) and Dual-sgRNA:Design 4 (**d**) rAAV8 vectors. Diagrams of references displaying notable construct domains are shown above each alignment. Regions upstream and downstream of the vector sequences (beyond ITRs) represent the bacterial plasmid backbone. The alignment summary shows read coverage. Read alignments displayed in IGV show matching nucleotides in gray. Nucleotide mismatches are depicted in green (A), red (T), blue (C), or brown (G); gaps are in black dashes; and inserts are in purple. Start and end position alignment counts (orange and blue traces, respectively) are displayed below each alignment. The percentages of read start and end peaks are displayed for prominent peaks. **e** Schematic diagram of the mouse *Hpd* gene. The first target site is represented by half arrowhead in red (gRNA-I) and the second target site is represented in green (gRNA-II). SMRT sequencing primers are also highlighted in gray, while UDiTaS primers are in orange.

to exhibit selective proliferation of edited cells, treatment with Dual-sgRNA:Designs 1 or 4 produced 50.5 ± 8.7% and 33.6 ± 6% segmental deletions between the two target sites, respectively (Fig. 2c). With the *Fah*$^{neo/neo}$ cohorts treated with each vector design, the mice analyzed before NTBC withdrawal exhibited 33.3% (Design 1) and 15.8% (Design 4) segmental deletion efficiencies. Three weeks following NTBC withdrawal and selective hepatocyte expansion, vector designs 1 and 4 yielded 35.7 ± 3.3% and 33.2 ± 4% segmental deletions, respectively. Surprisingly, inversions were detected at lower frequencies than previously demonstrated (Fig. 2c)[31,63]. In addition to segmental deletions, significant frequencies of small indels were observed at the two target sites (Fig. 2d). Vector fragment integration at the DSB sites were detected at a frequency of 2.8 ± 0.7% (Design 1) and 2.4 ± 0.8% (Design 4) in C57BL/6 mice, at 2.3% (Design 1) and 1.5% (Design 4) before NTBC withdrawal, and 3.5 ± 0.5% (Design 1) and 2.7 ± 0.5% (Design 4) post-NTBC withdrawal (Fig. 2e).

For the Dual-sgRNA:Design 4 AAV cohort, we used Tn5 tagmentation-based uni-directional targeted sequencing (UDi-TaS) analyses[31,63] to complement the vector editing profiles obtained by SMRT sequencing. Given that UDiTaS and SMRT sequencing have different locus-specific priming requirements (one or two primer binding sites, respectively) at different distances (nearer or farther, respectively) from the Cas9 target sites (Fig. 1e), they can capture distinct subsets of editing outcomes. When compared with SMRT sequencing, UDiTaS indicated lower frequencies of segmental deletion but higher frequencies of inversion and vector integration (Fig. 2f). Despite these quantitative differences, both readouts revealed efficient in vivo activity of the Dual-sgRNA:Design 4 AAV vector.

We extensively characterized genome-wide mRNA levels using mRNA extracted from mouse livers by mRNA sequencing. Our data revealed that efficient segmental deletion in recovered *Fah*$^{neo/neo}$ cohorts treated with each dual-sgRNA vector design yielded *Hpd* mRNA levels that were lower than those observed in healthy, PBS-treated wild-type mice (Fig. 2g, h). *Hpd* mRNA levels were also strongly reduced (relative to PBS-treated C57BL/6 mice) in PBS-injected *Fah*$^{neo/neo}$ animals, presumably as a secondary consequence of extensive liver damage[72]. *Hpd* mRNA levels were significantly decreased in C57BL/6 mice treated with either of the dual-sgRNA vectors (Fig. 2g), reflecting the expected results of Nme2Cas9 editing. Globally, 4792 and 5027 genes were significantly differentially expressed in *Fah*$^{neo/neo}$ mice after treatment with Dual-sgRNA:Design 1 and 4 vectors, respectively (FDR ≤5%, relative to PBS treatment). Differentially expressed genes were enriched within similar gene ontology categories across the two dual-sgRNA treatments, with an enrichment of genes in metabolic processes, regulation of endopeptidase activity, and response to external stimulus (Supplementary Tables). Hierarchical cluster analyses revealed a clear distinction between the untreated mice and treated cohorts that cluster in similar

patterns to the healthy C57BL/6 mice (Supplementary Fig. 2). In both cohorts of mice, there was a decrease in the inclusion of exons 3 and 4 in mature *Hpd* mRNA transcripts (Supplementary Fig. 2), with both exons 3 and 4 being skipped in 4.2–5.5% of transcripts (compared to 0.08% of transcripts in PBS-treated mice). More strikingly, due to the proximity of gRNA-II to the splice donor site (Fig. 1e), exon 4 is excluded in 86.7–87.5% of *Hpd* mRNAs in dual-sgRNA treated mice (compared to 0.33% of transcripts in PBS-treated mice) (Supplementary Fig. 2). Consistent with the observed prevalence of segmental deletions, inversions, skipped exons, and other perturbations (Fig. 2c, d, f), our protein analyses showed efficient knockdown of HPD in the livers of the edited *Fah*$^{neo/neo}$ and C57BL/6 mice by Western blot (Fig. 2i) and immunohistochemistry (Fig. 2j and Supplementary Fig. 2). Finally, we examined the accuracy of Nme2Cas9 at *Hpd* of gRNA-I and gRNA-II sites. We used CRISPRseek[73] to identify the four nearest-cognate PAM-matched sites as potential off-target sites, which we then analyzed for off-target editing by amplicon deep sequencing. We observed no significant indel frequencies at any of these sites, confirming the high fidelity of Nme2Cas9 in vivo[24,29] (Supplementary Fig. 7). These data demonstrate that Dual-sgRNA:Designs 1 and 4 are effective vectors for inducing segmental DNA deletions in vivo.

**Design of self-eliminating rAAV:HDR vectors.** The majority of rAAV-based applications to repair pathogenic mutations in vivo rely on a two-vector system for the delivery of Cas9 in one rAAV and the HDR donor (along with the guide) in another[40,74,75]. Although this strategy allows control over dosing at different donor:effector ratios, it also requires higher overall vector doses, which might be toxic to the patient. Furthermore, the lack of simultaneous transduction with both vectors in a percentage of cells will limit editing efficiencies. Previously, HDR-based precision editing platforms involving simultaneous delivery of Cas9, sgRNA, and donor DNA in a single vehicle generally used adenoviral vector[76] or RNP[77,78] delivery, neither of which can reach all cell types of clinical interest. To extend single-vehicle HDR-based gene editing to AAV vectors that are within the standard limits of efficient packaging (~4.9 kb) with well-characterized capsids, we engineered rAAV backbones containing Nme2Cas9, sgRNA, and a <500-nt donor DNA for in vivo testing of HDR editing.

To examine the best placement of the donor DNA within the rAAV backbone, we first created two designs in which the donor was placed near the 3′-ITR (design A) or in the middle of the backbone (design B) (Fig. 3a). We also created designs C and D, which respectively differ from A and B only in the presence of Nme2Cas9 sgRNA target sites flanking the donor (Fig. 3a). The presence of Cas9 target sites that flank an HDR donor has previously been shown to increase HDR efficiencies in cell culture after plasmid transfection[79]. These rAAV:HDR:cleaved designs also have the potential to enable self-elimination of the AAV:Nme2Cas9 vector, limiting or preventing long-term Cas9

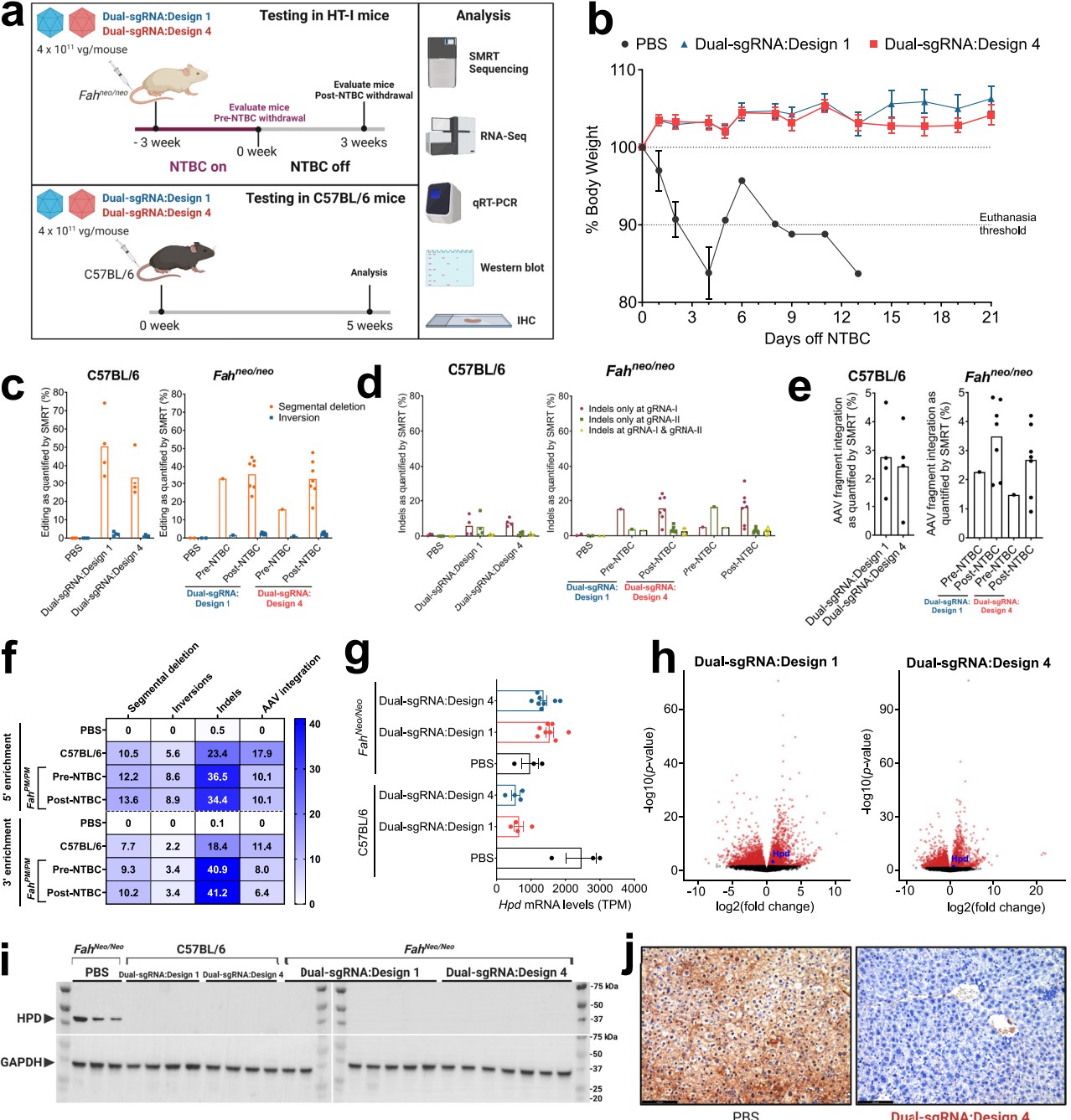

**Fig. 2 In vivo gene editing using dual-sgRNA rAAV:Nme2Cas9 vectors. a** In vivo experimental plan to validate rAAV:Nme2Cas9 Dual-sgRNA:Designs 1 and 4 to disrupt the *Hpd* gene by tail vein injections of AAV8 vectors in adult *Fah^neo/neo* and C57BL/6 mice. **b** Complete rescue of body weights after Dual-sgRNA:Designs 1 and 4 in *Fah^neo/neo* mice. **c** Quantification of editing events in C57BL/6 (left) and *Fah^neo/neo* (right) mice by SMRT sequencing analysis showing efficiencies of segmental deletion (orange) and inversion (blue) outcomes. **d** Bar graph showing the percentages of indels recorded in full-length (3.6 kb), UMI-corrected SMRT reads after *Hpd* editing by AAV8 delivery of Dual-sgRNA:Designs 1 and 4 in C57BL/6 (left) and *Fah^neo/neo* (right) mice. The plot indicates indels recorded only at gRNA-I (in red), indels only at gRNA-II (in dark green), and indels at both gRNA-I and gRNA-II (yellow) as measured by SMRT sequence analysis. **e** rAAV fragment integration as detected by SMRT sequencing analysis. **f** Quantitation of editing events in UMI-corrected UDiTaS analysis reads after *Hpd* editing by AAV8 delivery of Dual-sgRNA:Design 4 in C57BL/6 and *Fah^neo/neo* mice showing mean efficiencies of segmental deletion, inversion, and AAV fragment integration. **g** mRNA levels from RNA-seq (transcripts per million) of *Hpd* wild-type mRNA in the livers of C57BL/6 mice and *Fah^neo/neo* mice. **h** Volcano plots showing differentially expressed genes with false discovery rate ≤5% after adjusting for multiple comparisons (red; analysis detailed in Methods) between Dual-sgRNA:Design 1 and PBS-treated *Fah^neo/neo* mice (left) and Dual-sgRNA:Design 4 and PBS-treated *Fah^neo/neo* mice (right). **i** Total HPD protein knockout as shown by anti-HPD Western blot using total protein collected from mouse liver homogenates. This analysis of samples from multiple mice was performed once. **j** Representative images of immunostaining for HPD in liver tissues in *Fah^neo/neo* mice injected with PBS (left) or Dual-sgRNA:Design 4 (right). Scale bar is 100 μm. All mice in the cohort were examined once by this method (see Supplementary Fig. 2). Data were presented as mean values ± s.e.m. Sample size in panels **b-g**: (*n* = 3 PBS-injected C57BL/6 mice; *n* = 4 Dual-sgRNA.Design 1 or 4 injected C57BL/6 mice; *n* = 1 in pre-NTBC *Fah^neo/neo* cohort; *n* = 7 in post-NTBC Dual-sgRNA.Design 1 or 4 injected *Fah^neo/neo* cohort).

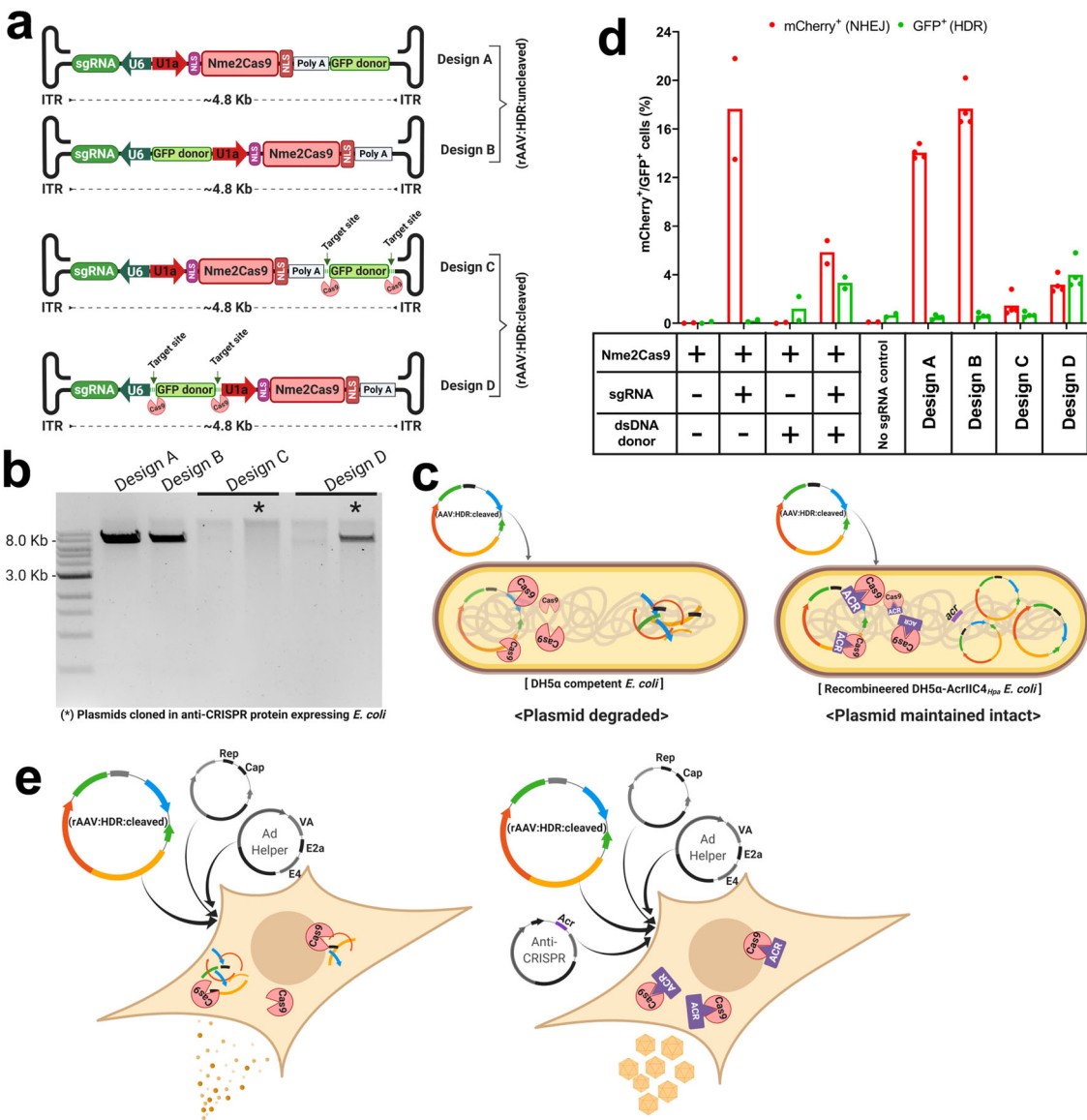

**Fig. 3 In vitro design and validation of all-in-one, self-inactivating rAAV:HDR vectors. a** Schematic of four different vector designs of rAAV:HDR constructs containing an sgRNA expression cassette, Nme2Cas9, and a donor DNA (<500 bp) with and without flanking sgRNA target sites. **b** Agarose gel electrophoresis (1% agarose) of linearized AAV:HDR plasmids. Asterisks indicate self-inactivated AAV:HDR plasmids that were cloned using anti-CRISPR-protein-expressing *E. coli* DH5α. DNA size marker size is 1 kb. Multiple plasmids were analyzed in this fashion three separate times, with consistent results. **c** Schematic of conventional *E. coli* DH5α and the recombined, anti-CRISPR-protein-expressing derivative used to successfully clone the self-targeting rAAV:HDR:cleaved plasmids. **d** Efficiencies of NHEJ and HDR events depicted as percentages of mCherry- and GFP-positive cells, respectively, obtained after transfection of AAV:Nme2Cas9:sgRNA and dsDNA GFP donor *in trans*), or of rAAV:HDR constructs in TLR-MCV1 HEK293T cells. The sample size represents independent transfection experiments (*n* = 2 of Nme2Cas9, sgRNA, and dsDNA donor *in trans* delivery; while *n* = 4 biological replicates of in *cis* delivery of AAV:HDR plasmids Designs A, B, C, and D). **e** Schematic of modified rAAV packaging transfection system using anti-CRISPR protein plasmid to block self-targeting by Nme2Cas9 expression in packaging cells during production.

expression in vivo[80–82] that could exacerbate off-target mutagenesis, genotoxicity, and immunogenicity. Previous attempts to build these self-eliminating AAV:Cas9 vectors relied on the delivery of a second AAV containing an sgRNA expression cassette to direct targeting of the Cas9-expressing AAV vector[82–84]. Such a system, although eliminating the AAV:Cas9 vector, is partially inadequate due to the continuous expression of AAV:sgRNA. By contrast, our all-in-one system averts the need for a second AAV vector and autonomously functions to correct the disease-causing mutation and eliminate itself.

The major hurdle in creating the rAAV:HDR:cleaved design C and D backbones was the degradation of the plasmids after transformation in *E. coli* competent cells due to leaky expression

of Nme2Cas9 and sgRNA in bacteria (Fig. 3b). To resolve this issue, we used λred phage recombineering technology[85] to integrate the *H. parainfluenzae* anti-CRISPR *acrIIc4Hpa* gene[86], under the control of its native promoter, into DH5α competent *E. coli* cells to inhibit Nme2Cas9 and prevent plasmid degradation (Fig. 3c). These plasmids were tested in a traffic light reporter (TLR) system[87], TLR-Multi-Cas-Variant 1 (MCV1), that has a disrupted GFP coding sequence followed by an out-of-frame mCherry cassette[88]. The reporter cassette was inserted as a single copy via lentiviral transduction in HEK293T cells. When a nuclease cleaves the insert that disrupts *eGFP*, some NHEJ-induced indels place the mCherry in a frame, inducing red fluorescence. By contrast, if an *eGFP* HDR donor is supplied and

used by the cell, the disrupted *eGFP* is repaired, inducing green fluorescence. We cloned an *eGFP* donor and an *eGFP*-insert-specific Nme2Cas9 sgRNA into the rAAV:HDR constructs to enable HDR to restore eGFP expression; in designs C and D, we also included sgRNA target sites flanking the donor. Our flow cytometry analyses show that design D resulted in eGFP signals that were comparable to those induced by a separate linear donor (generated by PCR) (Fig. 3d). For reasons that are unclear, yields of the design C plasmid remained low and we did not study it further. We chose design D and its non-self-inactivating analog (design B) (Supplementary Note) for in vivo testing and validation.

**In vivo validation of rAAV:HDR vectors in HT-I mice**. To examine the therapeutic potential of our all-in-one rAAV:HDR systems, we designed two vectors: rAAV:HDR:uncleaved (from design B) and the self-eliminating rAAV:HDR:cleaved (from design D) (Fig. 3a). For these studies we used HT-I mice that contain a point mutation in the last nt of exon 8 of the *Fah* gene[69] ($Fah^{PM/PM}$), leading to exon skipping that disables expression of the FAH protein[89]. Both vector designs contain Nme2Cas9, sgRNA targeting *Fah*, and a 358 bp HDR donor (including homology arms) to correct the G-to-A point mutation in *Fah* exon 8 in HT-I $Fah^{PM/PM}$ mice. The HDR donor sequence also included PAM mutations (CC > TT, within intron 8) at the target site to prevent cleavage of the donor fragment or of the HDR-repaired *Fah* locus. In addition, we created a negative control vector expressing an sgRNA with a non-cognate spacer sequence (rAAV:FahDonor:ncSpacer) and another with a non-cognate donor (rAAV:ncDonor:FahSpacer) (Fig. 4a). Neither of these negative control vectors is capable of self-targeting. These four ~4.8-kb vectors were packaged by the triple-transfection method for rAAV8 packaging[67]. Packaging of the rAAV:HDR:cleaved vector was modified to include a fourth plasmid expressing AcrIIC4$_{Hpa}$ to prevent the self-elimination of the rAAV:HDR:cleaved plasmid during production (Fig. 3e).

For in vivo evaluation, $8 \times 10^{11}$ gc of each vector was injected into adult HT-I $Fah^{PM/PM}$ mice via tail vein. Mice were maintained on NTBC for 5 weeks and then monitored for body weight loss after NTBC withdrawal (Fig. 4b). Mice injected with PBS, rAAV:FahDonor:ncSpacer, or rAAV:ncDonor:FahSpacer showed the expected gradual body weight loss and were euthanized. Strikingly, cohorts injected with rAAV:HDR:uncleaved and rAAV:HDR:cleaved initially lost weight but recovered and stayed healthy for 41 days (Fig. 4c). Some mice in the rAAV:HDR:cleaved and -uncleaved cohorts were put on NTBC-containing water temporarily to promote initial recovery.

To determine the efficiency of genome editing, we performed a targeted deep sequencing analysis to measure the level of genome modification at the *Fah* locus. HDR efficiency, calculated by co-conversion of all three edited positions [A..CC to G..TT], in pre-NTBC cohorts was very low (<0.1%); accurate quantifications were difficult to attain because of the inherent error rate of NGS, as well as the polyploid nature of hepatocytes and the presence of DNA from other nonparenchymal cells that are not efficiently transduced by AAV8[89] (Supplementary Fig. 3). HDR-edited reads were significantly increased to 4.4 ± 0.9% and 4.7 ± 0.4% in the rAAV:HDR:uncleaved and rAAV:HDR:cleaved cohorts, respectively 6 weeks after NTBC withdrawal. The relative increase over apparent editing efficiency in the wild-type cohort was likely due to the selective growth advantage of the successfully edited *Fah*+ cells after NTBC withdrawal (Fig. 4d and Supplementary Fig. 3).

We also performed immunohistochemistry analyses of liver tissues to examine the extent of liver recovery by FAH protein expression. Before drug withdrawal, we detected a marked

increase in the number of FAH+ cells in both rAAV:HDR cohorts (Fig. 4e and Supplementary 3). Interestingly, we observed through amplicon sequencing analysis that there were sporadic FAH+ cells in mice injected with rAAV:ncDonor:FahSpacer as a result of mutations that likely result from NHEJ repair events at the adjacent Nme2Cas9-catalyzed DSB (Supplementary Fig. 3). Additionally, those cells were likely expanded after NTBC withdrawal, although the starting population was apparently insufficient to allow the mice to recover from the disease phenotype in the absence of NTBC augmentation (Fig. 4e and Supplementary Fig. 3). A greater number of corrected FAH+ cells were observed in the rAAV:HDR cohorts before drug withdrawal, supporting eventual recovery (Fig. 4e and Supplementary Fig. 3). mRNA sequencing analyses showed that effective gene repair post-NTBC withdrawal was associated with significant increases in the level of *Fah* mRNA in both rAAV:HDR cohorts (relative to PBS, ncDonor, and ncSpacer controls, Fig. 4f and Supplementary Figs. 3 and 4). RT-PCR analysis with primers spanning exons 5 to 9 of *Fah* showed that the increased levels of *Fah* mRNA is also associated with exon 8 inclusion (Supplementary Fig. 3).

As expected by body weights, editing efficiencies, and FAH expression, HT-I mice treated with rAAV:HDR vectors exhibited striking reductions in liver damage as indicated by the significant reduction in serum aspartate aminotransferase (AST) and alanine aminotransferase (ALT) levels (Supplementary Fig. 3), as well as normalization of hepatocyte histology (Supplementary Fig. 3). Globally, 8827 and 8170 genes were differentially expressed in HT-I mice treated with rAAV:HDR:uncleaved and rAAV:HDR:-cleaved vectors, respectively (FDR ≤5%, relative to treatment with rAAV:ncDonor:FahSpacer vector; Supplementary Fig. 4). Differentially expressed genes were enriched among gene ontology categories involved in small molecule metabolic processes (Supplementary Table and Supplementary Fig. 4). Importantly, 54 genes were significantly differentially expressed between the rAAV:HDR:uncleaved and rAAV:HDR:cleaved vector treatments ($p \le 5\%$, Supplementary Fig. 4) in the treated mice after NTBC withdrawal, indicating distinct responses to targeting by the two vectors. RNA expression of several known oncogenes and tumor suppressors were included among the differences between untreated versus treated mice groups (Supplementary Fig. 4).

Previous efforts to create self-targeting rAAV:Cas9 vectors involved dual-vector systems[82–84]. Our all-in-one system necessarily involves co-expression of Nme2Cas9 and sgRNA in the same cell, enabling the targeting of engineered sites flanking the *Fah* HDR donor (Fig. 4a). Quantitative PCR analysis indicated that before NTBC withdrawal, the number of rAAV copies in liver samples did not differ significantly between the cleaved and uncleaved cohorts (Supplementary Fig. 5). However, in the post-NTBC withdrawal cohort where mice were maintained for a total of 11 weeks, animals treated with the rAAV:HDR:cleaved vector contained significantly fewer rAAV copies than the rAAV:H-DR:uncleaved mouse cohorts (Fig. 4g). This reduction in rAAV episomal numbers was not associated with a significant decrease in steady-state levels of Nme2Cas9 mRNA as measured by mRNA sequencing (Supplementary Fig. 5). To understand the basis for the apparently modest degree of self-inactivation, we amplified the junction outside of the donor region in the rAAV backbone using primers that are complementary to U1a and U6 promoter regions using total DNA from the mouse liver (Supplementary Fig. 5). Agarose gel electrophoresis revealed a ~500 bp amplicon in all cohorts corresponding to the vector backbone in the presence of the *Fah* or non-cognate donor (Supplementary Fig. 5). The rAAV:HDR:cleaved cohort also showed an additional band at ~140 bp, consistent in size with an amplicon from which the donor had been excised (Supplementary Fig. 5). We performed NGS analysis to confirm the identity

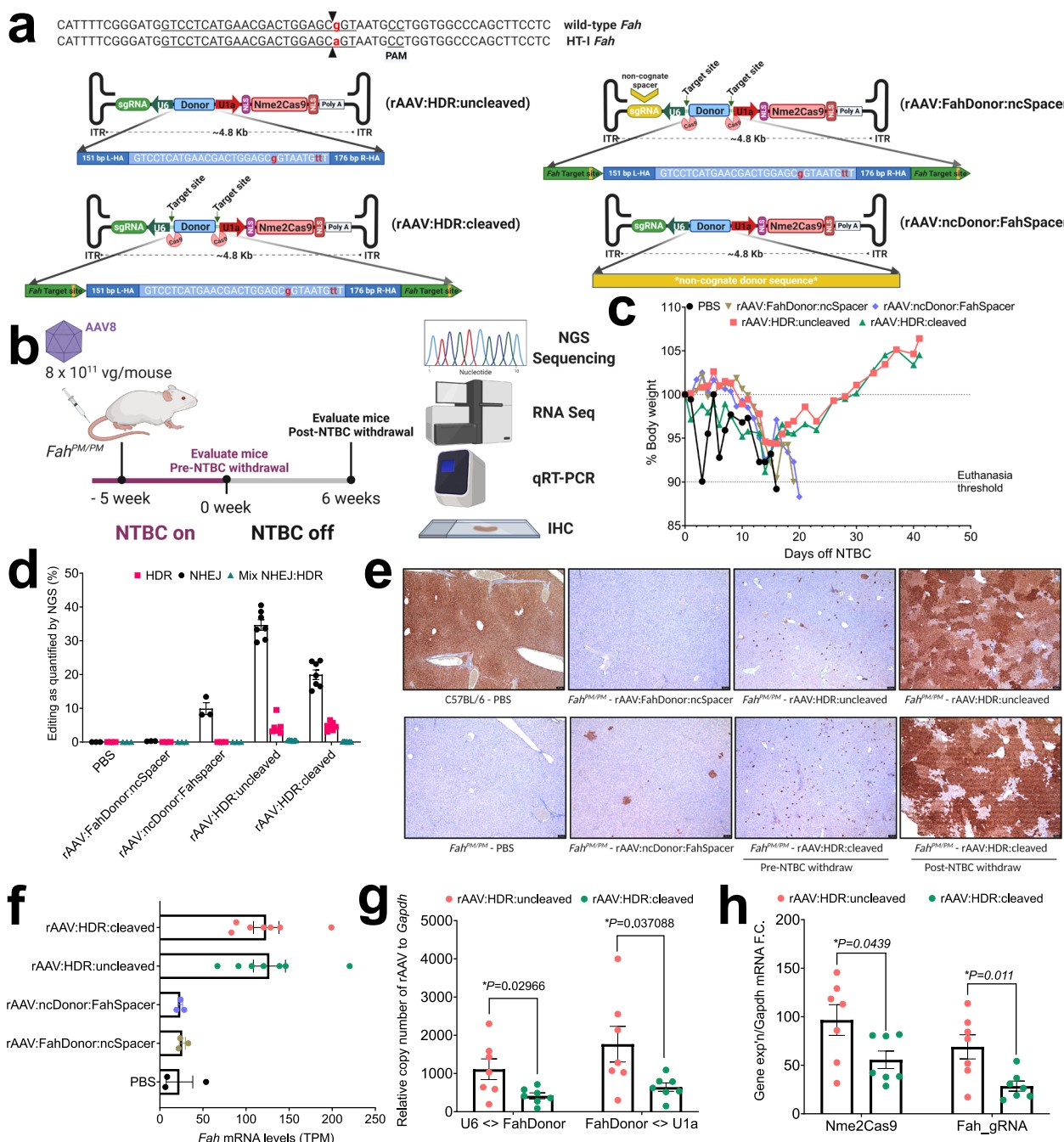

**Fig. 4 rAAV:HDR vectors rescue liver disease phenotypes in a mouse model of HT-I. a** Schematic of rAAV:HDR:uncleaved and -cleaved vectors (left) and control non-cognate donor or spacer vectors (right) to correct the *Fah* point mutation in *Fah*^PM/PM^ mice (top). **b** In vivo experimental regimen to correct the *Fah* mutation by injection of AAV8 through the tail vein in adult *Fah*^PM/PM^ mice. **c** Complete rescue of average body weight loss in *Fah*^PM/PM^ mice in treated cohorts injected with rAAV:HDR:uncleaved (red) and -cleaved (green), as compared to control cohorts injected with PBS (black), rAAV:FahDonor:ncSpacer (brown) or rAAV:ncDonor:FahSpacer (blue). Data shown represent the average body weight of each cohort. **d** Bar graph showing the percentage distribution of NHEJ, HDR, and imprecise NHEJ:HDR mix at *Fah* in livers of mice 6 weeks after NTBC withdrawal, as measured by NGS sequencing of PCR amplicons from genomic DNA. **e** Representative images of immunostaining for FAH in liver tissues of negative control and treated cohorts, as indicated below each panel. Scale bar, 100 μm. All mice in each cohort were examined once by this method (see Supplementary Fig. 3). **f** Levels of *Fah* mRNA from RNA-seq (transcripts per million) in the livers of *Fah*^PM/PM^ mice. **g** rAAV copy numbers in HT-I mouse liver tissues. rAAV copies are significantly reduced in rAAV:HDR:cleaved cohorts after NTBC withdrawal using the [U6<>FahDonor] primer pair (left) and [FahDonor<>U1a] primer pair (right). **h** qRT-PCR analyses with total RNA showing significantly reduced Nme2Cas9 mRNA and sgRNA expression in the rAAV:HDR:cleaved cohort after NTBC withdrawal. Data were presented as mean values ± s.e.m. Sample sizes in panels **c**, **d**, **f**–**h**: ($n = 3$ in PBS, rAAV:FahDonor:ncSpacer, rAAV:ncDonor:FahSpacer and pre-NTBC withdrawal rAAV:HDR:cleaved and -uncleaved cohorts; $n = 7$ in post-NTBC withdrawal rAAV:HDR:cleaved and -uncleaved cohorts). *p* values are calculated using Student's *t*-test (two-sided).

of this amplicon and found that indeed this amplicon is formed after the religation of the free ends of the rAAV:HDR:cleaved vector after *Fah* donor is excised, leading to the formation of vector genomes that may continuously express Nme2Cas9 mRNA and sgRNA (Supplementary Fig. 5). Although there was no reduction in steady-state Nme2Cas9 mRNA, total Nme2Cas9 mRNA, and sgRNA transcripts decreased over time, as measured by quantitative RT-PCR with random hexamer primers capturing both nascent and mature mRNAs, suggesting that the rAAV:HDR:cleaved platform can partially reduce the unwanted long-term expression of Nme2Cas9 (Fig. 4h and Supplementary Fig. 5). Additionally, we observed that the detected level of rAAV fragments integrated at the Cas9 DSB *Fah* site is significantly lower in the rAAV:HDR:cleaved cohort compared to rAAV:HDR:uncleaved (Supplementary Fig. 5). Finally, we examined the NGS data for homology-independent targeted integration (HITI)[90] of the donor by alignment of junction reads of *Fah* gene and donor sequence at the Nme2Cas9 DSB site. We detected a minimal number of aligned reads at the *Fah* locus (Supplementary Fig. 5), with all of them containing mutations in the integrated donor, indicating a low efficiency of HITI. Nme2Cas9 produced no detectable editing at predicted off-target sites as measured by amplicon sequencing analyses with DNA from mouse liver (Supplementary Fig. 7). All of these data suggest that the self-eliminating rAAV:HDR:cleaved vector is a more therapeutically viable option, as it can function in vivo to repair DNA mutations while reducing post-editing rAAV genome burden and Nme2Cas9 expression over time.

**In vivo validation of rAAV:HDR vectors in MPS-I mice.** The selective expansion of correctly edited cells in HT-I mice enabled therapeutic benefit even when editing efficiencies were very low. We sought a second disease model in which editing is not known to induce selective clonal expansion, to determine whether HDR efficiencies with our all-in-one rAAV system could still reach therapeutic thresholds. We chose MPS-I mice, which suffer from an accumulation of glycosaminoglycans (GAGs) in lysosomes. These mice carry a G-to-A mutation in the gene encoding the α-L-iduronidase (IDUA) enzyme, converting the Trp392 codon into a stop codon. Previous reports showed that IDUA enzyme activity as low as 0.2% of normal levels can provide therapeutic benefit in MPS-I patients[91,92], largely due to uptake of secreted IDUA by uncorrected cells. We engineered two *Idua*-targeting vectors analogous to those used above (rAAV:HDR:uncleaved and rAAV:HDR:cleaved) containing Nme2Cas9, a sgRNA targeting *Idua*, and a 272-bp donor (including homology arms) to correct the disease-causing G-to-A point mutation. Because the pathogenic mutation resides within an exonic protein-coding sequence, a PAM mutation is not feasible without altering the amino acid sequence. As an alternative approach, we incorporated two silent mutations (A..C > G..T) in the seed sequence of the spacer to prevent cleavage of the donor (Fig. 5a). Retargeting of the edited allele by Nme2Cas9 would be further inhibited by the third noncomplementary seed residue at the HDR-corrected position. For the negative control vector, we generated an rAAV backbone analogous to rAAV:HDR:uncleaved, but containing a non-cognate sgRNA spacer (rAAV:IduaDonor:ncSpacer) (Fig. 5a). All three vectors were packaged into AAV9 capsids, and cohorts of neonate MPS-I mice were injected with $4 \times 10^{11}$ gc of each vector via a facial vein (Fig. 5b). Liver tissue lysates were collected and IDUA enzymatic activities were measured. We observed significant increases in IDUA activity to $3.0 \pm 0.03\%$ and $2.3 \pm 0.2\%$ of wild-type levels in rAAV:HDR:uncleaved and rAAV:HDR:cleaved cohorts, respectively (Fig. 5c). This increase was above the therapeutic threshold required for effective

reduction of GAG accumulation in the liver (Fig. 5d). We also examined the activity of additional enzymatic markers that are commonly associated with MPS-I phenotypes in mice[93]. We found that the level of β-D-glucuronidase significantly decreased after treatment, while the level of D-hexosaminidase and lysosome-associated membrane glycoprotein 1 (LAMP-1) remained unchanged (Fig. 5e and Supplementary Fig. 6).

Targeted amplicon sequencing analysis of total liver DNA confirmed the corrected "G--T-G" pattern at the *Idua* locus in rAAV:HDR:uncleaved ($4.5 \pm 0.49\%$) and rAAV:HDR:cleaved ($3.3 \pm 0.43\%$) cohorts (Fig. 5f). Via qRT-PCR assays we also observed significant increases in *Idua* mRNA in the liver (Fig. 5g), likely due to reductions in nonsense-mediated mRNA decay of the edited transcripts. Other organs exhibited lower but notable levels of HDR (Supplementary Fig. 6).

To examine the self-cleaving ability of rAAV:HDR:cleaved vectors, quantitative PCR assays were designed using four primer pairs, including two that span the ends of the donor (where the cleavage sites reside in the self-targeting version) (Supplementary Fig. 6). Unlike rAAV:HDR:uncleaved cohorts, mice injected with the rAAV:HDR:cleaved vector had significantly lower qPCR signals in the liver (Fig. 5h), while Nme2Cas9 mRNA and sgRNA transcripts were modestly reduced, as measured by qRT-PCR (Fig. 5i, j). To understand the reason behind the continuous expression of Nme2Cas9 and sgRNA, we examined the rAAV:HDR:cleaved cohort for the religation of the rAAV:HDR:cleaved vector after donor excision. Our agarose gel electrophoresis and NGS data confirmed that some AAV genomes are religated making smaller rAAV backbones lacking the donor, but with sgRNA and Nme2Cas9 expression cassettes intact (Supplementary Fig. 6). The frequency of rAAV fragment integration at the *Idua* target site (as detected by NGS sequencing) was minimal, with significantly fewer insertion events detected with the rAAV:HDR:cleaved vector (Supplementary Fig. 6). We examined the NGS data for HITI events at this locus by aligning to the *Idua* donor junction reads at the Nme2Cas9 DSB site. We also observed a population of aligned reads although all of those reads contained multiple mutations giving no indication for HITI-based integration of the donor at this locus (Supplementary Fig. 6). Since these mice were injected as neonates, Nme2Cas9-specific humoral immune responses did not develop in MPS-I mice, while adult-treated HT-I and healthy C57BL/6 mice treated with rAAV:Nme2Cas9 targeting the *Rosa26* locus[24] generated readily detectable α-Nme2Cas9 IgG1 antibodies with no apparent difference between rAAV:HDR:cleaved and rAAV:HDR:uncleaved cohorts (Supplementary Fig. 6). Nme2Cas9 produced no detectable editing at predicted off-target sites of *Idua* as measured by NGS analysis in mice liver (Supplementary Fig. 7). In conclusion, rAAV:HDR vectors have the potential to correct inherited disease mutations in vivo, while enabling the therapeutic benefits of longer-term self-elimination to minimize genotoxic and off-target effects, as well as possible cellular immune responses against transduced tissues.

## Discussion

The tremendous potential of precision genome editing in vivo to repair pathogenic mutations still faces significant translational hurdles. AAV vectors for Cas9 genome editing exhibit great promise for some tissues and are already in clinical trials for NHEJ-based edits (ClinicalTrials.gov; identifier: NCT03872479). However, rAAV delivery systems for precise genomic rewriting via HDR or base editing have required multiple vectors[40–47]. The extreme rAAV dosages needed for multi-vector strategies impose safety difficulties in clinical applications[49–51]. Here, we have developed single-vector systems with standard rAAV

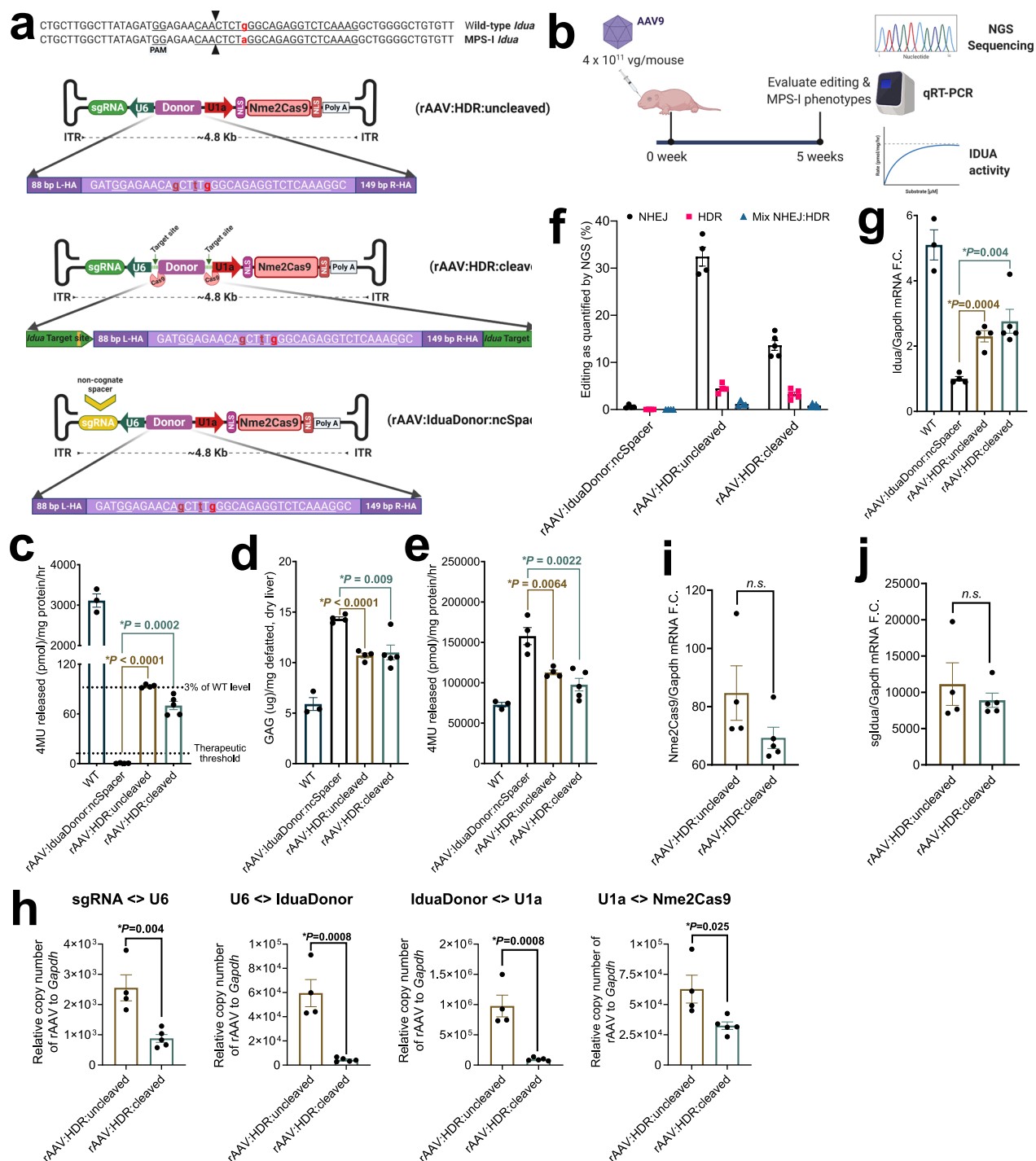

**Fig. 5 Reduced clinical disease manifestation in a mouse model of mucopolysaccharidosis type I treated with rAAV:HDR vectors. a** Schematic of rAAV:HDR:uncleaved, -cleaved, and control non-cognate spacer vectors to correct the *Idua* point mutation in MPS-I *Idua*[W392X] mice. **b** In vivo experimental regimen to correct the *Idua* mutation by injection of rAAV9 through the facial vein in neonate MPS-I mice. **c** IDUA specific activity in liver lysates of healthy mice compared to rAAV:IduaDonor:ncSpacer mice (negative control) and treated rAAV:HDR uncleaved/cleaved cohorts. The dashed lines indicate 0.2% (therapeutic threshold) and 3% of the IDUA activity detected in lysates from WT mice. **d** Level of glycosaminoglycan (GAG) accumulation in the livers of the same mice shown in (**c**). **e** β-D-glucuronidase specific activity in liver lysates of healthy mice compared to rAAV:IduaDonor:ncSpacer mice (negative control) and treated rAAV:HDR uncleaved/cleaved cohorts. **f** Bar graph showing the percentage of NHEJ, HDR, and imprecise NHEJ:HDR mix at *Idua* in the liver as measured by NGS sequencing of PCR amplicons from genomic DNA. **g** qRT-PCR data showing increase in the relative *Idua* mRNA in the liver normalized to *Gapdh* mRNA. **h** AAV copy numbers in MPS-I mouse liver tissues using the [sgRNA<>U6] [U6<>IduaDonor], [IduaDonor<>U1a], and [U1a<>Nme2Cas9] primer pairs. **i, j** Animals treated with rAAV:HDR:cleaved vectors exhibit reduced Nme2Cas9 mRNA (i) and sgRNA (j) expression levels as measured by qRT-PCR, though reductions do not reach statistical significance. Data in panels **c–h** are presented as mean values ± s.e.m. (*n* = 3 mice in the WT cohort, *n* = 4 mice in the rAAV:IduaDonor:ncSpacer and rAAV:HDR:uncleaved cohorts, and *n* = 5 mice in the rAAV:HDR:uncleaved cohort). *p* values are calculated using Student's *t*-test (two-sided).

serotypes and cargo limits for precise Nme2Cas9 editing via HDR. Nme2Cas9 is well-suited for this strategy because it is compact, active in adult mammals in vivo, highly accurate within eukaryotic cells, and has a broad targeting range due to its modest (dinucleotide) PAM requirement[24,53]. We further employed an Acr protein during cloning and vector production to enable the inclusion of self-targeting sites that can facilitate vector self-elimination over time, once the Nme2Cas9/sgRNA complex has accumulated to functional levels. The use of Acr proteins to protect otherwise self-inactivating genome editing vectors was reported previously for adenovirus vectors[94] but to our knowledge has not been described for the more commonly employed rAAV. The ever-expanding roster of anti-CRISPRs against different genome editing effectors[54,55] suggests that this strategy will be useful far beyond Nme2Cas9. We validated our all-in-one rAAV HDR platform in vivo in two disease models in mice (HT-I and MPS-I), and in both cases editing efficiencies reached therapeutic thresholds. Our ability to fit an HDR donor into a single-vector system along with Nme2Cas9-NLS and sgRNA transgenes was enabled by minimizing the latter components and their regulatory elements. We also generated all-in-one vectors encoding two guides and demonstrated that this dual-sgRNA all-in-one system was effective in vivo for inducing segmental deletions in HT-I mice. In the latter case, sgRNA cassette placement and orientation was a critical factor in vector packaging efficiency, as revealed by a nanopore sequencing approach that we introduce here to characterize the genomes of rAAV after packaging to assess integrity. Our results with these analyses highlight the importance of careful consideration of the cargo's secondary structure while engineering rAAV vectors. The two successful designs involved precise targeting of sgRNAs to exon boundaries, resulting in an unexpected diversification of *Hpd* mRNA splicing events through unknown mechanisms that merit further study. Although our nanopore sequencing data show that dual-sgRNA design 4 produced more homogeneous vector genomes than produced by design 1, our SMRT sequencing data confirms that both vectors efficiently induce segmental deletions.

Although therapeutic thresholds were reached by the all-in-one HDR system in vivo for the two preclinical disease models that we tested, HDR efficiencies were nonetheless low in absolute terms. This is not particularly surprising because of several factors: (i) the lengths of the homology arms needed for HDR are constrained by rAAV vector cargo limits; (ii) donor DNA cannot be delivered in excess of the guide- and effector-expressing vector; and (iii) many mammalian cells and tissues in vivo are quiescent or terminally differentiated, leading to a general loss of HDR capacity[16]. In its current form, our single-AAV-vector HDR approach would therefore be limited to applications where successfully edited cells have a proliferation advantage (as with HT-I), or where low levels of correction suffice for clinical benefit (as with MPS-I). Nonetheless, in such cases, the reduction in vector dose permitted by the use of a single-vector system could be crucial for the safety profile of precision therapeutic editing. A previous report also described an all-in-one rAAV system for HDR-based editing and applied it to HT-I mice[52]. However, this method used a VP2-deleted capsid system that enabled larger cargoes (>5 kb) but appeared to package inefficiently[52]. We report here that standard rAAV vector platforms that accommodate <4.9 kb genomes can be used for single-vector HDR in vivo.

In addition to reductions in vector dose, safety prospects are also potentially improved by our system's implementation of self-cleavage ability, enabled by anti-CRISPR inhibition of Cas9 activity during cloning and packaging. Our qPCR assays suggest modest degrees of rAAV genome depletion over 5- to 11-week editing regimens, although self-targeting clearly occurred as revealed by the detection of indels and segmental deletions in the vector genomes. Whether self-elimination increases over longer time courses will require more extended follow-up studies; indeed, while this paper was in revision, another report described a more substantial reduction in response to self-targeting after 24 weeks[95]. One potential improvement in future self-targeting systems will be to place the sgRNA target sites within the expression cassettes—e.g., in nonessential promoter and UTR regions—to disrupt expression in religated vectors, for instance by promoter deletion. We did not observe detectable increases in vector fragment insertion into the chromosomal target site with the self-inactivating vector. We designed our self-targeting system to have the added benefit of generating a linear HDR donor, in part to test the hypothesis that such a configuration would improve HDR efficiency over that of the uncleaved HDR vector. We obtained no clear evidence that donor excision capacity improves HDR frequencies in vivo, though the potential long-term benefits of vector elimination still apply, even if HDR enhancement via donor cleavage cannot be detected in other physiological contexts in the future.

In conclusion, we have established advanced viral delivery platforms for in vivo genome editing that enable precision editing via HDR, using single-vector systems that are compatible with well-characterized rAAV backbones and serotypes. We have further developed the means to enable cloning, packaging, and use of rAAV vectors endowed with self-targeting capabilities. These systems improve the balance between the competing imperatives of improving vector capabilities (e.g., enabling precise editing via HDR) and limiting vector dose to maximize safety, reduce cost, and mitigate production bottlenecks. As additional routes toward increased HDR efficiency emerge[96,97], this could enable single-vector HDR-based rAAV delivery systems to address clinical indications that demand higher levels of precise gene correction, perhaps including those that are difficult or impossible to achieve via base editing or prime editing.

## Methods

**Hereditary tyrosinemia type I mice**. All animal study procedures were approved by the Institutional Animal Care and Use Committee (IACUC) at the University of Massachusetts Medical School. HT-I mice were kindly provided by Dr. M. Grompe and maintained on C57 background for the *Fah^{PM/PM}* strain, and on a 129 background for *Fah^{neo/neo}*. Mice were kept on 10 mg/L 2-(2-nitro-4-tri-fluoromethylbenzoyl)-1,3-cyclohexanedione (NTBC; Sigma-Aldrich, Cat. No. PHR1731-1G) in drinking water when indicated. Male and female mice (8 to 15 weeks old) were used, and their body weights were recorded every 1–3 days. Moribund mice with more than 10% body weight loss were humanely euthanized, and sera and liver tissues were harvested for genome editing analyses. Tissues were pulverized in liquid nitrogen and aliquots of tissue powder (~30 mg) were used to extract total DNA [using the DNeasy Blood and Tissue Kit (Qiagen), according to the manufacturer's protocol], as well as total RNA and total protein.

**Mucopolysaccharidosis type I mice**. Homozygous MPS-I *Idua^{W392X}* mice[93] were purchased from the Jackson Laboratory (Stock No. 017681) and used to breed neonatal *Idua^{W392X}* pups for rAAV injections. Postnatal day 1 mice were injected with $4 \times 10^{11}$ gc of rAAV:HDR:uncleaved, rAAV:HDR:cleaved, or control vectors via the facial vein. Five weeks after injection, blood was collected via cardiac punch, and mice were transcardially perfused with ice-cold PBS. Tissues were immediately dissected, snap-frozen in liquid nitrogen, and stored at −80 °C. Both female and male animals were used. All animal procedures were reviewed and approved by the IACUC at the University of Massachusetts Medical School and performed in compliance with all relevant ethical regulations.

**Cloning of rAAV plasmids**. All rAAV vectors were based on our previously published rAAV plasmid [Nme2Cas9_AAV; Addgene plasmid #119924] design[24], with modifications to increase the cargo capacity (Supplementary Fig. 1). Human-codon-optimized *Neisseria meningitidis* DE10444 Cas9 (Nme2Cas9) is under the expression of a *U1a* promoter, with one SV40 nuclear localization signal (NLS) on the N terminus and one nucleoplasmin NLS on the C terminus, followed by a short poly(A) signal[59]. Shortened sgRNA Nme.sgRNA-121 was synthesized as a gblock for cloning. Target spacer sequences were inserted by digesting the plasmid with the BspQI restriction enzyme, and the annealed spacer sequence carrying

appropriate BspQI overhang sequences were ligated. This plasmid is available on Addgene (pEJS1089: mini-AAV:sgRNA.Nme2Cas9; Addgene # 159536).

**Cloning of dual-guide rAAV:Nme2Cas9 plasmids.** A second *U6* promoter (from human *U6* for designs 1, 2, and 4; and from mouse *U6* for design 2) with a Nme.sgRNA-121 expression cassette was generated as a gblock (IDT) and cloned into the minimized rAAV:Nme2Cas9 plasmid to create dual-sgRNA designs 1–4 (Fig. 1a). The two configurations that enabled efficient packaging of full-length vector genomes (designs 1 and 4) are available on Addgene (pEJS1099: Dual-sgRNA:Design 4, Addgene #159537; pEJS1096: Dual-sgRNA:Design 1, Addgene # 159538). Oligonucleotides with spacer sequences for target genes were inserted into the sgRNA cassette by ligation into the BspQI and BsmBI cloning sites.

**Cloning of rAAV:HDR plasmids.** Donor DNA fragments with target sites (rAAV:HDR:cleaved) or without target sites (rAAV:HDR:uncleaved) were synthesized as gBlocks (IDT) and cloned into the minimized Nme2Cas9+sgRNA AAV backbone plasmid following digestion of the latter with SalI restriction enzyme. The rAAV:HDR:uncleaved plasmids were cloned in NEB 5-alpha Competent *E. coli* cells (Catalog # C2987H), while rAAV:HDR:cleaved plasmids were cloned into the DH5α-AcrIIC4_{Hpa} *E. coli* strain that we constructed. Oligonucleotides with spacer sequences for target genes were inserted into the sgRNA cassette by ligation into the BspQI cloning sites.

**Creating *E. coli* DH5 cells stably expressing anti-CRISPR protein DH5α-AcrIIC4_{Hpa}.** Cells were created by λRed-promoted PCR-mediated recombination[85]. Briefly, *acrIIc4_{Hpa}*, its promoter, and Kan^R cassette were cloned from plasmid pUC57mini-AcrIIC4_{Hpa} (generously provided by Dr. Alan Davidson's lab at the University of Toronto) into the pKIKOarsBKm (Addgene #46766) plasmid by restriction digestion and ligation into the *arsB* locus of DH5α. Cells harboring plasmid pKM208 (Addgene #13077) were cultured at 30 °C to an OD of 0.35, induced with 1 mM IPTG, and harvested at an OD of 0.80. Cells were prepared for electroporation (washing with 10% glycerol) and transformed with 500 ng of gel-purified PCR product (*acrIIc4_{Hpa}*, its promoter, and Kan^R). After incubation at 37 °C overnight in kan-LB media, colonies were screened by PCR for the *acr* gene and the expected junction of the recombineering product with the *arsB* gene. Correct clones were passaged at 42 °C to eliminate the pKM208 plasmid.

The rAAV:HDR:cleaved plasmids were linearized with BspQI restriction enzyme and spacer sequences with compatible overhangs were ligated into 100 ng of plasmid backbone using ElectroLigase® (NEB M0369) following the manufacturer's protocol. For electro-transformation, 2 μL of the ligation reaction was added to 40 μL of DH5α-AcrIIC4_{Hpa} *E. coli* and electroporated at 2500 V, 200 Ω, 25 μF, in a 2 mm gap cuvette using Gene Pulser Xcell Electroporation Systems [Bio-Rad #1652660].

**Vector production.** Packaging of AAV vectors was done at the Viral Vector Core of the Horae Gene Therapy Center at the University of Massachusetts Medical School[67]. Constructs for HT-I studies were packaged in AAV8 capsids, whereas constructs for MPS-I studies were packaged in AAV9 capsids. A plasmid expressing anti-CRISPR protein (pEJS581; pCSDest2-AcrIIC4_{Hpa}-FLAG-NLS; Addgene # 113436) was included in the triple-transfection packaging process to maintain intact rAAV:HDR:cleaved plasmids during production. Vector titers were determined by droplet digital PCR, and by gel electrophoresis followed by silver staining.

**Cell culture.** HEK293T cells harboring TLR-Multi-Cas-Variant 1 (TLR-MCV1)[88], as well as Neuro2a cells (ATCC CCL-131), were cultured in Dulbecco's Modified Eagle Media (DMEM, Thermo Fisher Scientific, Cat. No. 11965084); while MOLT-3 cells were cultured in RPMI-1640 medium. Both media were supplemented with 10% FBS (Thermo Fisher Scientific, Cat. No. 10438034) and 1% Penicillin/Streptomycin (GIBCO). Cells were incubated at a 37 °C incubator with 5% $CO_2$.

**Plasmid transfection.** HEK293T cells and TLR-MCV1 cells were transfected using PolyFect Transfection Reagent (Qiagen, Cat No. 301105) with 400 ng of plasmid in a 24-well plate according to the manufacturer's protocol. Neuro2a cells were transfected with 1000 ng of plasmid using Lipofectamine 3000 Reagent (Thermo Fisher Scientific, Cat. No. L3000015) following the manufacturer's protocol. FACS analysis for the TLR-MCV1 cells were performed 3 days post-transfection using a Miltenyi MACSQuant VYB flow cytometer. HEK293T and Neuro2a cells were harvested 3 days post-transfection and genomic DNA was extracted using a DNeasy Blood and Tissue kit (Qiagen) according to the manufacturer's protocol.

**Ribonucleoprotein nucleofection.** Nme2Cas9 protein was expressed in Rosetta (DE3) cells and purified using a Ni$^{2+}$-NTA agarose column (QIAGEN)[24]. Imidazole was used to elute the bound protein followed by dialysis into storage buffer [20 mM HEPES-NaOH (pH 7.5), 1 mM DTT, 250 mM NaCl]. Wild-type (wt) gRNA was in vitro transcribed using the HiScribe T7 High Yield RNA Synthesis Kit (New England Biolabs) according to the manufacturer's protocol. Truncated guide (Nme.sgRNA-100) was chemically modified with 2′-*O*-methyl analogs and 3′-phosphorothioates in 5′ and 3′ terminal RNA residues and was synthesized by

Synthego (Redwood City, CA, USA). RNP complex was electroporated into HEK293T or MOLT-3 cells using the Neon transfection system. Briefly, 40 picomoles (HEK293T and MOLT-3 cells) of 3xNLS-Nme2Cas9 and 50 picomoles (HEK293T and MOLT-3 cells) of T7-transcribed or chemically synthesized sgRNA was assembled in buffer R along with 100,000 HEK293T cells or 200,000 MOLT-3 cells and electroporated using 10 μL tips. Electroporation parameters (voltage, width, number of pulses) were 1150 V, 20 ms, 2 pulses for HEK293T cells; 1600 V, 10 ms, 3 pulses for MOLT-3 cells.

**TIDE analysis, amplicon sequencing, and indel analysis.** PCR was carried out with TIDE or amplicon sequencing primers as shown in the Supplementary Table using High Fidelity 2x PCR Master Mix (New England Biolabs). PCR products were purified using the DNA Clean & Concentrator-100 (Zymo, Cat. No. D4029) and sent for Sanger sequencing using a TIDE forward primer (Supplementary Table). Indel readouts were obtained using the TIDE web tool (https://tide-calculator.nki.nl/)[27]. For amplicon sequencing, PCR was carried out using amplicon sequencing primers shown in the Supplementary Table. Libraries were sequenced for 600 cycles (300/300 PE) on a MiSeq (*Fah* locus), and for 300 cycles (SE) on a MiniSeq (*Idua* locus). FASTQ files were trimmed with *Cutadapt* (Galaxy Version 1.16.6)[98] and analyzed using CRISPResso 2.0.40 with parameters specifying guide, amplicon, and expected HDR sequences. A no-guide sample was used as the negative control to determine the background. AAV integration was quantified using BWA-MEM (Galaxy Version 0.7.17.1) and samtools to align sequenced amplicons files to the *Idua* locus and AAV vector sequence[99]. We searched amplicon reads for HITI[90] using CRISPResso 2.0.40, providing unedited, homology-repaired, sense HITI, and antisense HITI reference alleles.

**Amplicon sequencing analysis of rAAV:HDR:cleaved religation after donor excision.** We designed an NGS library using genomic DNA from liver tissues. Each primer was complementary to the U6 and U1a promoters flanking the donor (Supplementary Table). The gene-specific primer contained UMIs and adapter overhangs to multiplex the library. Samples were amplified by linear amplification (98 °C for 40 s, followed by 12 cycles of 98 °C for 45 s, 64 °C for 20 s, and 72 °C for 18 s) using the forward primers. A second PCR amplification was performed using a universal primer and a gene-specific reverse primer at 98 °C for 30 s, followed by 30 cycles of 98 °C for 10 s, 62 °C for 20 s, and 72 °C for 18 s, and extended at 72 °C for 5 m. PCR products were diluted (1:100) and amplified using Illumina barcoded primers at 98 °C for 30 s, followed by 30 cycles of 98 °C for 15 s, 65 °C for 20 s, and 72 °C for 75 s, and extended at 72 °C for 12 m. Equimolar amounts of the products were combined together and purified by gel extraction then AMPure PB beads.

We used CRISPResso 2.0.40 to align the reads to the intact AAV sequence, cleaved AAV sequence, and AAV sequence where the sequence between the cut sites is inverted. The aligned reads either mapped perfectly or contained small insertions or deletions as shown.

**CRISPRseek off-target prediction and editing analysis.** Bioconductor package CRISPRseek (Version 1.32.0) was used to predict off-target sites of Nme2Cas9 using mouse reference (mm10)[24]. The following parameters were sgRNA.size = 24, PAM = "NNNNCC," PAM.size = 6, RNA.PAM.pattern = "NNNNCN," and candidate off-target sites of a maximum of six mismatches. Top predicted off-targets with N4CC PAM and matching seed sequence to the on-target were analyzed by NGS using Genomic DNA liver tissue from mice. Illumina amplicon sequencing library was prepared using a two-step PCR protocol. Briefly, regions of predicted off-target loci (around 250 bp) were amplified from genomic DNAs using forward and reverse primers that contain Illumina adapter sequences (Supplementary Table). The PCR products were amplified using forward and reverse primers that contain unique barcode sequences. Products with the correct size were extracted from agarose gel and then purified by Ampure XP beads. The concentration of the final purified library was determined using Qubit (High Sensitivity DNA assay). The integrity of the library was confirmed by Agilent Tapestation using the Agilent High Sensitivity D1000 ScreenTape kit. The library was sequenced on an Illumina Miniseq platform according to the manufacturer's instructions using Miniseq Mid Output Kit (300 cycles). Sequencing reads were demultiplexed on the Miniseq and CRISPResso 2.0.40 was used to align the reads and quantify editing efficiencies.

**Vector library preparation and nanopore sequencing.** Viral vector DNAs were extracted by phenol-chloroform extraction and ethanol precipitation[66]. Samples (<1 mg) were subjected to library preparations for Oxford Nanopore Technologies (ONT) sequencing, following the protocol for 1D Native barcoding of genomic DNA, using the Native Barcoding Expansion 1–12 (PCR-free) (EXP-NBD103) and Ligation Sequencing Kit (SQK-LSK108) components. Libraries were then multiplexed and sequenced on an ONT MinION instrument (Flow Cell R9.4.1, FLO-MIN106D).

**Nanopore sequencing data analysis.** MinKNOW software (v19.10.1) was used for base calling, adapter trimming, and demultiplexing. Sequencing reads were aligned to the appropriate reference sequences by BWA-MEM[100] within the Galaxy web-based interface[101–104]. Alignments were visualized using Integrative Genome Viewer (IGV, version 2.6.3)[105] with soft-clipping displayed. Start and end positions

of aligned reads were defined with BEDTools (version 2.27.0)[106,107]. The abundances of start and end positions of all mapped reads were tabulated and plotted using GraphPad Prism 8.4.3.

**SMRT sequencing.** Amplicon libraries were constructed using locus-specific primers as shown in the Supplementary Table using Q5® High-Fidelity DNA Polymerase (New England Biolabs). One primer of each pair contains an 8-nucleotide UMI, and both contain adapter overhangs to multiplex the library using Bar. Univ. F/R Primers Plate-96v2 (Pacbio part no. 101-629-100). Forward and reverse primers were 1.1 and 1.9 kb from gRNA-I and gRNA-II Cas9 DSB sites, respectively. Each sample was amplified by linear amplification (98 °C for 40 s, followed by 10 cycles of 98 °C for 30 s, 64 °C for 20 s, and 72 °C for 75 s) using the forward primers. A second PCR amplification was performed using a universal PacBio primer and a gene-specific reverse primer at 98 °C for 30 s, followed by 30 cycles of 98 °C for 15 s, 64 °C for 20 s, and 72 °C for 75 s, and extended at 72 °C for 5 m. PCR products were diluted (1:100) and amplified using PacBio barcoded primers at 98 °C for 30 s, followed by 30 cycles of 98 °C for 15 s, 64 °C for 20 s, and 72 °C for 75 s, and extended at 72 °C for 12 m. Equimolar amounts of the products were combined together and purified by AMPure PB beads (PacBio, part no. 100-265-900) following the manufacturer's recommendation. The libraries were sequenced on a Pacific Biosciences Sequel II Instrument in a 10-h collection SMRTCell™. A circular consensus sequence (ccs) reads generated by SMRT Link (smrtlink/8.0.0.79519) using default parameters in fastq format were processed and analyzed using the Galaxy web platform at usegalaxy.org (20.09.rc1)[108] with custom workflows, unless specified. Briefly, reads were demultiplexed using *FASTQ/A Barcode Splitter* (Galaxy Version 1.0.1) and converted to fasta to calculate and count read lengths within each library (Supplementary Fig. 2). To assign reads that represent segmental deletions, inversions, or indel events, reads were first re-IDed by appending each sequence identifier with the UMIs using fastp with (-U). Duplicated UMIs were removed from the analysis. Reads with mapping qualities >20 were then uniquely aligned using BWA-MEM 0.7.17 to reference sequences representing the wild-type loci and the predicted sequences following segmental deletion or inversion. The counts for reads aligning to the specified references were displayed as a percentage of all mapped reads. To categorize reads that mapped to the wild-type reference as either bearing indel events or unedited genomes at indicated target loci, reads were trimmed to ±100 nt surrounding each target PAM by *Cutadapt* (Galaxy Version 1.16.6). Trimmed reads were then converted to fasta and clustered by *UCLUST* (usearch v7.0.1090)[109], a centroid-based clustering algorithm. Singlets were discarded and treated as reads bearing sequencing errors. Clustered reads using size reporting (-sizeout) were categorized into unedited and edited groups at each target site or both sites for dual-guide vectors and displayed as a percentage of all reads mapping to the respective wild-type sequence. To assess target loci that harbor vector genome integration events, fastq reads were mapped to the relevant vector genome reference and expressed as a percentage of all reads.

**UDiTaS sequencing.** Genomic DNA was quantified using the Qubit dsDNA BR kit according to the manufacturer's instructions. About 200 ng of genomic DNA was tagmented with a Tn5 transposon loaded with custom oligos according to the protocol from ref. [63] (Supplementary Table). An initial round of nested PCR was done for one target with a gene-specific primer (Hpd_2.22_nest_FWD) paired with an i5-specific primer for ten cycles. This was followed by purification with Ampure beads at 0.9X and a second round with an interior gene-specific primer (Hpd_2.22_FWD or Hpd_4.10_REV) for 12 cycles. The PCR products were again purified with Ampure beads at 0.9X and a final PCR of 15 cycles was performed to add the Illumina i7 adapter and barcodes. Amplicons were double size-selected with Ampure beads at 0.5X/0.35X to yield a final size distribution of 200–1000 bp. Paired-end sequencing was performed on an Illumina Miniseq for 2 × 150 cycles.

Transposon-mediated target enrichment and sequencing at the *Hpd* locus after dual-guide editing was analyzed using the workflow outlined by Giannoukos et al.[63] (UDiTaS v1.0) with modifications allowing for MiniSeq platform compatibility. Briefly, paired-end, dual-indexed libraries were demultiplexed using bcl2fastq (Illumina), masking UMIs (I2 nts 1–9). Next, un-demultiplexed UMIs were stored in a single FASTQ using bcl2fastq (–create-fastq-for-index-reads parameter), masking R1, I1, I2 nts 10–17, R2. Demultiplexed R1 and R2 FASTQs were paired with their corresponding UMIs using the fastq-pair[110] hash lookup tool. Gzipped FASTQs were placed into individual UDiTaS-compatible sample directories (i.e., "parent/samplename/fastq_files/"). Per UDiTaS source code, FASTQs were named: samplename_R1.fastq.gz; samplename_R2.fastq.gz; samplename_umi.fastq.gz. Reference FASTA and 2-bit files were obtained from the UCSC table browser[111]. Zero-order primer and sgRNA coordinates were obtained from reference FASTA using the twobitreader python package. Sample metadata were saved as sample_info.csv in the "parent/" directory. Analysis was executed inside the umasstr/UDiTaS docker container, skipping demultiplexing.

**RNA sequencing.** Total RNA was isolated from liver tissues using TRIzol reagent and separated by chloroform. RNA-seq library preparation was performed with 1 ug of total RNA per sample using the TruSeq Stranded mRNA Library Prep kit (Illumina, 20020595) following the manufacturer's protocol. Libraries were made for the following mice: three PBS-treated *Fah^neo/neo*, eight dual-sgRNA Design 1

treated *Fah^neo/neo*, eight dual-sgRNA Design 4 treated *Fah^neo/neo*, three healthy C57BL/6, four dual-sgRNA Design 1 treated C57BL/6, four dual-sgRNA Design 4 treated C57BL/6, three PBS-treated *Fah^PM/PM*, three rAAV:FahDonor:ncSpacer treated *Fah^PM/PM*, three rAAV:ncDonor:FahSpacer treated *Fah^PM/PM*, ten rAAV:HDR:uncleaved treated *Fah^PM/PM*, and ten rAAV:HDR:cleaved treated *Fah^PM/PM*. Libraries were sequenced on an Illumina NextSeq 500 with single-end 75 nucleotide reads and an average of 19.88 million reads. All downstream analyses were performed with the Ensembl GRCm38.p6 (GCA_000001635.8) *Mus musculus* genome adapted to include the Nme2Cas9 mRNA sequence. Transcripts Per Million (TPM) were calculated with alignment-free quantification using Kallisto v0.45.0. Reads were mapped to the mm10 genome using STAR v2.7.5a in chimeric mode to identify chimeric junction reads and HTSeq v.0.10.0 was used to quantify read counts for each gene. Differential expression analyses were performed with DESeq2 release 3.11 after filtering out genes with less than ten read counts across all samples in the comparison. Gene ontology analyses were conducted with a custom script to test for enrichment of functional annotations among significantly differentially expressed genes, in order to avoid significant gene ontology terms with overlapping gene sets. Specifically, the script uses gene ontology annotation databases from the Gene Ontology Consortium to perform iterative enrichment analyses and *p* values are computed using a Fisher-exact test and then corrected using a Benjamini–Hochberg multiple test correction.

Gene expression heatmaps were created using the DolphinNext RNA-seq pipeline[112] (revision 4), including Illumina adapter removal using the built-in trimmomatic software (v0.39). Default parameters were used to process raw reads using RSEM (v1.3.1) and STAR to map the reads to a reference mouse reference (mouse_mm10_refseq). Levels of differentially expressed genes were assessed using DESeq2 software (v1.28.1) using the following parameters: fit type: "parametric", betraPrior = FALSE, test type = LRT" (Likelihood Ratio Test) and shrinkage = None. Alpha (padj) was set to 0.05 and a minimum fold change was set to 2. Heatmaps of (all) or (selected) up- and down-regulated genes were plotted with MRN normalization.

**HPD western blotting.** Liver tissue fractions were ground, resuspended in 200 µL of RIPA lysis buffer, and allowed to lyse for 15 min on ice. Measurements of total protein content was determined by Pierce™ BCA Protein Assay Kit (Thermo-Scientific) following the manufacturer's protocol. A total of 25 µg of protein was loaded onto a 4–20% Mini-PROTEAN® TGX™ Precast Gel (Bio-Rad) and transferred onto a PVDF membrane. After blocking with 5% Blocking-Grade Blocker solution (Bio-Rad) for 2 h at room temperature, membranes were incubated with rabbit anti-HPD (Sigma HPA038322, 1:600) overnight at 4 °C. Membranes were washed three times in TBST and incubated with horseradish peroxidase (HRP)-conjugated goat anti-rabbit (Bio-Rad 1,706,515, 1:10,000) secondary antibodies for 2 h at room temperature. The membranes were washed four times in TBST and visualized with Clarity™ western ECL substrate (Bio-Rad) using a Bio-Rad ChemiDoc MP imaging system. Membranes were stripped using Restore™ Western Blot Stripping Buffer (Cat. number: 21059) following the manufacturer's protocol and blocked with 5% Blocking-Grade Blocker solution (Bio-Rad) for 2 h at room temperature. The membranes were incubated with rabbit anti-GAPDH (Abcam ab9485, 1:2,000) overnight at 4 °C, washed three times in TBST, and incubated with HRP-conjugated goat anti-rabbit (1:4,000) secondary antibodies for 2 h at room temperature. The membranes were washed four times in TBST and visualized with Clarity™ western ECL substrate (Bio-Rad) as above.

**Histological analysis.** Liver tissues were harvested, and sections were placed in biopsy cassettes and fixed in 10% buffered formalin overnight before changing into 70% ethanol solution. FAH immunohistochemistry (IHC) was performed using an anti-FAH antibody (Abcam, Cat. No. ab81087; dilution 1:400), whereas HPD IHC was performed using an anti-HPD antibody (Sigma HPA038322; dilution 1:30). IHC and hematoxylin and eosin (H&E) staining were performed by the Morphology Core at the University of Massachusetts Medical School following standard procedures.

**Reverse transcription PCR.** Total RNA was isolated from liver tissues using TRIzol reagent and separated by chloroform. Reverse transcription reactions were carried out using SuperScript™ III First-Strand Synthesis System (Thermo Fisher Scientific; 18080051) following the manufacturer's protocol. The complementary DNA was used to quantify *Fah* gene expression in a qPCR reaction using primers shown in the Supplementary Table using PowerUp SYBR Green Master Mix (Applied Biosystems, Cat. No. A25742). Data were collected using QuantStudio Design and Analysis Desktop Software V1.5.1 on a QuantStudio 3 Real-Time PCR System (Applied biosystems). All reactions were normalized to *gapdh* and relative quantification in gene expression was determined using the $2^{-\Delta\Delta Ct}$ method on Microsoft Excel Version 1902 (Build 11328.20644). PCR products were resolved by agarose gel electrophoresis.

**Quantitative PCR for rAAV copy number.** Genomic DNA from mouse tissue was used to quantify rAAV copy number using primers shown in the Supplementary Table using PowerUp SYBR Green Master Mix (Applied Biosystems, Cat. No. A25742). Data were collected using QuantStudio Design and Analysis Desktop

Software V1.5.1 on a QuantStudio 3 Real-Time PCR System (Applied biosystems). All reactions were normalized to *gapdh* control and relative quantification of rAAV copies was determined using the $2^{-\Delta\Delta Ct}$ method relative to PBS-injected cohorts on Microsoft Excel Version 1902 (Build 11328.20644).

**Serum aspartate transaminase (AST) and alanine transaminase (ALT) assays**. Blood was collected from mice by cardiac puncture immediately before euthanasia, and sera were isolated using a serum separator (BD, Cat. No. 365967) and stored at −80 °C. AST and ALT levels were determined using a Cobas Clinical Chemistry Analyzer at the MMPC-University of Massachusetts Medical School Analytical Core (RRID:SCR_015365). Two mice in the PBS- and rAAV:ncDonor:FahSpacer negative control HT-I cohorts were hemolyzed, so we were able to measure ALT/AST for only one mouse. In the rest of the groups, data were presented as mean values ± s.e.m. (*n* = 3–7 mice per group).

**Iduronidase, β-d-glucuronidase, and β-d-hexosaminidase activity assays**. The three assays were done similarly using the indicated substrates[93,113,114]. Tissues were homogenized in ice-cold T-PER protein extraction reagent (Thermo Fisher Scientific, Cat. No. 78510) with protease inhibitor (Roche, Cat. No. 4693159001) using TissueLyser II (Qiagen). Quantification of total protein concentration was assessed by the bicinchoninic acid (BCA) method (Pierce, Cat. No. 23225). No more than 80 μg of total protein was used in enzymatic reactions (100 μL of total reaction volume), which includes sodium formate buffer, pH 3.5 (130 mM), D-saccharic acid 1,4-lactone monohydrate (0.42 mg/mL, Sigma-Aldrich, Cat. No. S0375), and 4MU-iduronic acid (0.12 mM, Santa Cruz Biotechnology, Cat. No. sc-220961). The reaction was incubated at 37 °C for 24–48 h and quenched with glycine buffer (pH 10.8). The fluorescence of released 4MU (excitation wavelength: 365 nm; emission wavelength: 450 nm) was detected using a fluorescence plate reader (BioTek) and compared against a standard curve generated using 4MU (Sigma-Aldrich, Cat. No. M1381). The iduronidase specific activity was calculated as 4MU released (pmole) per milligram of total protein per hour. The β-D-glucuronidase assay was done using 4MU-β-D-glucuronide (1 mM, Sigma, Cat. No. M9130) substrate, while the β-D-hexosaminidase activity assay was done using 4MU-β-D-hexosaminide (4 mM, Sigma, Cat. No. M2133) substrate and incubation time was set to 30 min.

**Glycosaminoglycan (GAG) assay**. Tissues were homogenized in a mixture of chloroform and methanol (2:1) using TissueLyser II (Qiagen) and dried in a Vacufuge (Eppendorf) to remove fat. The dried and defatted tissue was weighed and digested using papain (Sigma-Aldrich, Cat. No. P3125) at 60 °C overnight. The supernatant was used in the Blyscan assay to quantify GAG content using chondroitin-4-sulfate as standard (Accurate Chemical, Cat. No. CLRB1000). Levels of GAG were calculated as GAGs (microgram) per milligram of dried, defatted tissue[114].

**LAMP-1 western blotting**. Liver tissues were homogenized in ice-cold T-PER protein extraction reagent (Thermo Fisher Scientific, Cat. No. 78510) with protease inhibitor (Roche, Cat. No. 4693159001) using TissueLyser II (Qiagen). The supernatant was used to quantify total protein concentration using the bicinchoninic acid (BCA) method (Pierce, Cat. No. 23225) and boiled with 4x Laemmli Sample Buffer (Bio-Rad, Cat. No. 1610747) at 95 °C for 10 min. Primary antibody rat anti-LAMP-1(BD Pharmingen, RUO - 553792) (1:2000 dilution) and secondary antibody LI-COR IRDye 680RD Goat Anti-Rat IgG (H + L) (LI-COR Biosciences, 926–68076) (1:5000 dilution) were used in western blot. The membrane was scanned with a LI-COR scanner (Odyssey). Western blot quantification analysis was performed with Image Studio Lite (LI-COR).

**Humoral α-Nme2Cas9 immune response**. A 96-well plate (Corning) was coated with 0.5 μg of recombinant Nme2Cas9 protein suspended in a 1× coating buffer (Bethyl E107). The plate was incubated for 12 h at 4 °C with shaking, then washed three times using 1x Wash Buffer (Bethyl E106). The plate was blocked with a blocking buffer (Bethyl E104) for 2 h at room temperature. After washing three times, diluted sera (1:40 in dilution buffer consisting of 1 L of TBS, 10 g of BSA, and 5 mL of 10% Tween 20) from mice injected with rAAV:Nme2Cas9 were added to each well in duplicates and incubated at 4 °C for 6 h. The plate was washed three times, and a goat anti-mouse HRP-conjugated antibody (100 μL; 1:100,000 dilution; Bethyl A90-105P) was added to each well. The plates were incubated at room temperature for 1 h, then washed four times. To detect immune responses, the plate was developed by adding 100 μL of TMB One Component HRP Substrate (Bethyl E102). After incubating in the dark for 30 min at room temperature, Stop Solution (Bethyl E115; 100 μL per well) was directly added. Absorbance was measured at 450 nm using a BioTek Synergy HT microplate reader and collected using the Gen5 software.

**Statistical analysis**. All data are presented as mean values ± s.e.m. *p* values are calculated using Student's *t*-test (two-sided) and one-way ANOVA using GraphPad Prism 8.4.3.

**Reporting Summary**. Further information on research design is available in the Nature Research Reporting Summary linked to this article.

## Data availability

All sequencing data that support the findings of this study have been deposited in the NIH Sequence Read Archive via BioProject PRJNA667456. Other source data are provided with this paper as a Source Data file. Source data are provided with this paper.

## Code availability

All analysis was performed using publicly available programs and the parameters indicated in the Methods section.

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

## Acknowledgements

We thank members of the Sontheimer, Gao, Wang, Xue, and Wolfe labs for helpful advice and discussions, and Alan Davidson for anti-CRISPR plasmid. We are grateful to the UMass Medical School Mouse Phenotyping Center, Viral Vector Core, Sequencing Core, Morphology Core and Animal Medicine staff for their expertize and assistance. We also thank Krishna Ghanta for his assistance with SMRT sequencing library design, Jordan Smith for help with animal handling, and Jun Xie for AAV packaging expertize. We thank Jay Nagpaul for building the umasstr/UDiTaS docker container and for his valuable contributions in modifying the UDiTaS analysis pipeline. Multiple figures of this manuscript were created with BioRender.com (2021). Support for this work was provided by the National Institute of Diabetes and Digestive and Kidney Diseases of the National Institutes of Health (F31DK120333 to R.I.), as well as NIH grants R01NS076991 and U19AI149646 to G.G., P01HL131471 and UG3HL147367 to G.G. and W.X., DP2HL137167 to W.X., R35GM133762 to A.A.P., R01GM125797 to E.J.S., and R01GM115911 and UG3TR0228 to S.A.W. and E.J.S. Additional support was provided by funds from SLC61A Connect to A.A.P. and by grants from the American Cancer Society (129056-RSG-16-093) and the Cystic Fibrosis Foundation to W.X.

## Author contributions

R.I. and E.J.S. conceived the study and designed experiments with the participation of all authors. R.I. engineered and tested plasmids and AAV vectors, processed mouse tissues, assembled sequencing libraries, and performed all Western blot, qRT-PCR, qPCR, tissue imaging, and immune response assays. A.M. designed and tested truncated sgRNAs and A.M. and R.I. validated truncated sgRNAs in cellular editing. A.M. and T.C.R. performed computational analyses of short-read amplicon libraries. P.W.L.T. analyzed SMRT long-read sequencing data. S.N. and P.W.L.T. analyzed vector DNA and optimized nanopore sequencing analysis. E.M. and S.M. prepared UDiTaS sequencing libraries and T.C.R. performed the computational analysis. Z.C. prepared NGS sequencing libraries for off-target analysis. J.W. handled MPS-I mice and performed IDUA and GAG assays. N.J. and E.S.K. prepared RNA-Seq libraries, N.J. and A.A.P. analyzed RNA-seq data and E.T. created gene expression heatmaps using DolphinNext. S.N. constructed E. coli strains expressing the anti-CRISPR protein. Y.C. and E.T. assisted with animal experiments. S.A.W., D.W., A.A.P., W.X., G.G., and E.J.S. oversaw experimental design and interpretation and all authors assisted in data interpretation. R.I. and E.J.S. wrote the manuscript and all authors edited the manuscript.

## Competing interests

The authors declare the following competing interests: A patent application has been filed on technologies related to this work by the University of Massachusetts Medical School [Names of inventors: Erik J. Sontheimer, Raed Ibraheim, Wen Xue, Aamir Mir, Alireza Edraki, Ildar Gainetdinov. Application number: 16/186,352. Status of application: Pending.] The following aspects of the manuscript are covered in the patent application: (1) Truncating of sgRNA sequence to create 121-nt sgRNA versions (Nme.sgRNA-121). (2) Truncating of sgRNA sequence to create 100-nt sgRNA versions (Nme.sgRNA-100). (3) Minimized all-in-one AAV.Nme2Cas9.sgRNA vector backbone to a packaging size of 4.4 Kb including its promoters and regulatory signals. (4) Dual-sgRNA AAV.NmeCas9 vector backbones. G.G. is a scientific co-founder of Voyager Therapeutics, Adrenas Therapeutics, and Aspa Therapeutics and holds equity in these companies. G.G. is an inventor on patents with potential royalties licensed to Voyager Therapeutics, Aspa Therapeutics, and other biopharmaceutical companies. E.J.S. is a co-founder and scientific advisor of Intellia Therapeutics. The remaining authors declare that the research was conducted in the absence of any commercial or financial relationships that could be construed as a potential conflict of interest. The authors declare no competing interests.
