## [Peer Review File · Nature Communications]

Reviewers' Comments:

Reviewer #1:

Remarks to the Author:

The study by Ibraheim et al is a comprehensive and well-written report describing the design and characterization of an all-in-one AAV vector system for genome editing via HDR in vivo. The authors develop NmeCas9 along with two guide RNAs in a single vector system, optimize positional effects within the AAV genome and carry out a robust analysis of the system in 3 mouse models with 2 editing techniques, two different high throughput sequencing methods to measure editing efficiency and establish phenotypic recovery supported by DNA, RNA and protein analysis. An all-in-one vector system <4.8kb is a significant step towards enabling efficient genome editing with CRISPR/Cas9 systems. Some significant concerns are noted below.

1. Data supporting the self-inactivating CRISPR vector appears to fall short of expectations/benefits claimed by the authors due to several reasons. It should be noted that the anti-crispr based design approach is elegant and addresses the challenges of propagating the self-eliminating plasmids in bacteria as well as packaging within AAV vectors. However, data in Fig 4/FigS5 shows reduction in genomes and mRNA/gRNA levels is only partial. Although not shown, it is likely that Cas9 protein expression is still detectable and functional at 5/11 week time intervals. The reasons behind lack of efficient self-inactivation are not clear. That longer time intervals might be required is proposed, but it is not evident why AAV episomes would pose a target for self-elimination at later time intervals over earlier. A related question that needs to be addressed is whether there is a dose threshold for self-elimination of genomes. In addition, no significant impact is observed on gene editing outcomes. Overall, the self-inactivating system does not appear to be very effective due to unknown reasons and the benefits are not readily apparent.
2. The authors claim that the NmeCas9 shows higher fidelity than other Cas9 systems, but off-target effects for gRNAs against specific targets described in the current study have not been analyzed.
3. The authors present GO analysis of the differentially expressed/impacted genes in the control vs treatment arms, it would be useful to know if any of those genes are tumor suppressor/oncogene related.

Reviewer #2:

Remarks to the Author:

In the study by Ibraheim et. al., the authors developed an all-in-one AAV vector containing Nme2Cas9/2-sgRNAs or Nme2Cas9/sgRNA/HDR through minimizing the Nme2Cas9 cassette and reducing sgRNA scaffold sequence. Furthermore, by introducing two target sites into the all-in-one plasmid, the AAV vector was able to achieve self-elimination after transduction. Through this single vector system, two strategies have been tested to be feasible for the amelioration of two genetic disorders in mouse model. This is a very hard work and an extensive evaluation of the performance of this system. The major contributions of this study are all-in-one vector system and self-elimination of AAV vector by employment of anti-CRISPR protein. However, these two points have been published in previous reports. Although this study may contribute the community of gene therapy, it lacks novelty in genome editing field.

Major concerns:

1. What is the advantage of dual-sgRNA AAV:Nme2Cas9 vector? Since at most circumstances, one efficient sgRNA is enough to disrupt a gene of interest. Two sgRNAs will increase off-target potential. What is the application scenario of these two sgRNA strategy for gene therapy? This issue should be discussed. Moreover, Cpf1 could be used for multiple sgRNA targeting since its crRNA is shorter and multiple crRNAs could be processed by Cpf1 transcribed by single U6 promoter.
2. In Figure 1b Dual-sgRNA design 4 produce much more full-length viral DNA than Dual-sgRNA design 1 (Judging from the band intensity and design 1 exhibited a strong band with similar size of the control—shortened single-sgRNA). However, this result doesn't match the in vivo segmental deletion efficiencies in Figure 2c where design 1 is better in both WT and Fahneo/neo model. Some explanation is needed because the choice of vector design is based entirely on the result of Figure 1b.
3. Using different PolIII promoter to decrease the sequence similarity is an alternative strategy to

reduce vector truncation. It is better to compare different strategies.

4. In figure 1 d and e, what is the total ratio of editing efficiency, including segmental deletion inversion and indel?

5. In figure 2, why the segmental deletion ratio of mice treated with Design 1 did not increase after NTBC withdraw? In figure 2d, why the indel rate of Design 4 is largely different pre- and post-NTBC? (Increased indels only at gRNA-1 but opposite at gRNA-II). Single deletion caused by sgRNA-I is enough to disrupt Hpd gene? However, in page 8 line 18 of the third paragraph, exon 4 is excluded in 86-87% of Hpd mRNA which is probably due to gRNA-II.

6. In Figure 3c the Nme2Cas9 expression level among design A, B, C D should be compared. An explanation is needed for the low cleavage performance of Design C.

7. Since AAV is a single stranded DNA virus, it suggests that ssDNA induces higher HDR rate than dsDNA in cells. Why two sgRNA released HDR template induced higher HDR efficiency? Moreover, two sgRNA released template could be inserted into the target site in the genome similar to HITI strategy (Nature. 2016, 540(7631):144-149). It should be analyzed.

8. The HDR efficiency is very low in both of the disease models even for single base repair. Due to the capacity of AAV vector, HDR template could not be extend much longer for insertion of an entire ORF. Thus, what is the advantage of this system compared to base editing which achieves much high base conversion.

9. In the experiment of curing the MPS-I model, the authors did not include the rAAV:ncDonor:IduaSpacer virus, which would help to exclude the NHEJ mediated read-through of the stop codon and restoring of the enzyme activity. Authors should check their NGS result to see whether there was any case like this.

10. In the MPS-1 correction assay, did the authors discovered any NHEJ mediated targeted integration? The NHEJ mediated targeted integration will also correct the point mutation but would introduce indels in the joint sites.

11. MPS-1 gene therapy is a novel model, but only enzyme activity was determined. More phenotypic studies should be presented to show the efficacy.

RESPONSE TO CRITIQUES

We are grateful to the reviewers for their critiques. Our revision adds new data (including nanopore sequencing analysis of vector genomes, molecular analysis of vector self-inactivation events, and expanded documentation of editing accuracy in vivo) and also updates the text in ways that address the stated concerns. Specific revisions are further detailed below and denoted in red font in the manuscript text. We believe that the manuscript has been significantly improved as a result of these efforts and we thank the referees for helping us strengthen the study.

Reviewer #1 (Remarks to the Author):

The study by Ibraheim et al is a comprehensive and well-written report describing the design and characterization of an all-in-one AAV vector system for genome editing via HDR in vivo. The authors develop Nme2Cas9 along with two guide RNAs in a single vector system, optimize positional effects within the AAV genome and carry out a robust analysis of the system in 3 mouse models with 2 editing techniques, two different high throughput sequencing methods to measure editing efficiency and establish phenotypic recovery supported by DNA, RNA and protein analysis. An all-in-one vector system <4.8kb is a significant step towards enabling efficient genome editing with CRISPR/Cas9 systems. Some significant concerns are noted below.

Response: First we would like to thank the reviewer for the helpful feedback and for sharing our enthusiasm about all-in-one HDR systems. We are grateful for the constructive comments regarding the various points and have addressed them experimentally and edited the manuscript to reflect those changes.

1. Data supporting the self-inactivating CRISPR vector appears to fall short of expectations/benefits claimed by the authors due to several reasons. It should be noted that the anti-crispr based design approach is elegant and addresses the challenges of propagating the self-eliminating plasmids in bacteria as well as packaging within AAV vectors. However, data in Fig 4/FigS5 shows reduction in genomes and mRNA/gRNA levels is only partial. Although not shown, it is likely that Cas9 protein expression is still detectable and functional at 5/11 week time intervals. The reasons behind lack of efficient self-inactivation are not clear. That longer time intervals might be required is proposed, but it is not evident why AAV episomes would pose a target for self-elimination at later time intervals over earlier. A related question that needs to be addressed is whether there is a dose threshold for self-elimination of genomes. In addition, no significant impact is observed on gene editing outcomes. Overall, the self-inactivating system does not appear to be very effective due to unknown reasons and the benefits are not readily apparent.

Response: We are thankful to the reviewer for highlighting the novelty of deploying the anti-CRISPR protein technology as a key to successful design and production of self-inactivating AAV vectors for gene editing. The reviewer raises a very important point regarding the efficiency of the self-inactivation mechanism of the rAAV:HDR:cleaved vector system. We share the reviewer's stance that longer (>11 weeks) time points might be required to reach efficient self-elimination levels of the vector, especially in light of another very recent manuscript [PMID 33718515] that noted ~20-fold self-targeting vector reduction after 24 weeks, also using qPCR as we did in our initial submission.

Our revision now incorporates deep sequencing-based analyses proving that self-targeting occurs (Supplementary Fig. 5f and Supplementary Fig. 6g), and also pointing towards strategies for next-generation self-targeting implementations that will more thoroughly disrupt effector

expression. First and foremost, using our 5-week and 11-week post-injection samples, we readily detect amplicons in which the HDR donor segment flanked by the two self-targeting Nme2Cas9 cleavage sites is deleted. This provides unambiguous evidence that vector cleavage is in fact occurring. The 2-3x decrease in qPCR signal at 5-11 weeks that we originally reported (Fig. 4g and 5h) with primers outside of the targeted region, along with the results described in PMID 33718515, also indicate that self-targeting can induce vector loss, especially at later time points. Nonetheless, perhaps not surprisingly, our results also demonstrate that dual-site self-cleavage can lead to end-joining repair events that yield indels and segmental deletions. Given the positioning of the cleavage sites at the ends of the HDR donor, segmental deletion and repair yields a shorter vector sequence that retains the sgRNA and Nme2Cas9 transcription units, explaining the modest effect on Nme2Cas9 expression originally noted. However, the clear implication is that cleavage site placement that disrupts transcription units (e.g. within promoters and UTRs) will have much stronger effects on Cas9 expression, e.g. due to promoter deletion. Our debut report on <4.8kb, self-targeting, all-in-one AAV vectors, and its molecular evidence for successful self-targeting and AAV vector genome reduction even after only 5 weeks, will therefore set the stage for follow-up reports where self-cleavage site positioning throughout the vector can be more systematically explored.

2. The authors claim that Nme2Cas9 shows higher fidelity than other Cas9 systems, but off-target effects for gRNAs against specific targets described in the current study have not been analyzed.

Response: We thank the reviewer for raising this important concern. Our previous work on Nme2Cas9 protein [PMID 30581144] showed empirically that it is a consistently high-accuracy nuclease as revealed by GUIDE-Seq analysis with numerous guides, as well as amplicon deep-sequencing. While our revision of the current manuscript was in progress, another report appeared [PMID 33992807] corroborating our earlier conclusion of consistently high editing accuracy with this platform. Nonetheless, in response to the reviewer's concern as it pertains to the specific guides used in this study, we conducted global off-target prediction analysis using CRISPRseek, identified the most closely matched genomic sites with NNNNCC PAMs, and assessed off-target indel frequencies by NGS using genomic DNA from livers of treated mice. As expected, no indels could be detected at these potential off-target sites for the *Hpd* gRNA-I, *Hpd* gRNA-II, *Fah* and *Idua* loci that we used in this study (Supplementary Fig. 7). These results reaffirm the established hyper-accuracy of Nme2Cas9. The inclusion of this analysis has strengthened our revision and we are grateful to the referee for raising this point.

3. The authors present GO analysis of the differentially expressed/impacted genes in the control vs treatment arms, it would be useful to know if any of those genes are tumor suppressor/oncogene related.

Response: We thank the reviewer for this very helpful comment. We have added this analysis to the manuscript (see Supplementary Fig. 4d).

Reviewer #2 (Remarks to the Author):

In the study by Ibraheim et. al., the authors developed an all-in-one AAV vector containing Nme2Cas9/2-sgRNAs or Nme2Cas9/sgRNA/HDR through minimizing the Nme2Cas9 cassette and reducing sgRNA scaffold sequence. Furthermore, by introducing two target sites into the all-in-one plasmid, the AAV vector was able to achieve self-elimination after transduction. Through this single vector system, two strategies have been tested to be feasible for the amelioration of two genetic disorders in mouse model. This is a very hard work and an extensive evaluation of the performance of this system. The major contributions of this study are all-in-one vector system and self-elimination of AAV vector by employment of anti-CRISPR

protein. However, these two points have been published in previous reports. Although this study may contribute the community of gene therapy, it lacks novelty in genome editing field.

Response: We are grateful to the reviewer for the time and effort devoted toward our manuscript. We respectfully disagree with the claim that our report lacks novelty. To the best of our knowledge, all-in-one AAV systems incorporating guide, donor and Cas9, using clinically validated, efficiently packaged capsids and vector backbones, have never been reported before for any Cas9, let alone for a high-accuracy, dinucleotide-PAM-targeting system like Nme2Cas9. Similarly, we are not aware of any report in which multiple-guide single AAV vectors have been employed in vivo for a hyper-accurate Cas9 with such broad targeting range. We are especially puzzled by the reviewer's claim -- unsupported by any specific references -- that "self-elimination of AAV vector by employment of anti-CRISPR protein" was published in previous reports. We believe that this claim is simply wrong. The strategy was reported once before for adenovirus vectors (PMID 31080846), but unlike AAV, adenovirus vectors have not won wide acceptance or use in the genome editing field. While our revision was in progress, another self-inactivating AAV manuscript appeared (PMID 33718515); the strategy (if any) used to prevent self-targeting during cloning and packaging is not well described in that study, but the use of anti-CRISPRs for this purpose is never mentioned.

Major concerns:

1. What is the advantage of dual-sgRNA AAV:Nme2Cas9 vector? Since at most circumstances, one efficient sgRNA is enough to disrupt a gene of interest. Two sgRNAs will increase off-target potential. What is the application scenario of these two sgRNA strategy for gene therapy? This issue should be discussed. Moreover, Cpf1 could be used for multiple sgRNA targeting since its crRNA is shorter and multiple crRNAs could be processed by Cpf1 transcribed by single U6 promoter.

Response: This comment suggests that the expert referee has forgotten many of the basic tenets and applications of genome editing strategy. Although ORF knockout by individual indels is indeed a common application of Cas9 genome editing, it is far from the only one, as the field's literature readily reveals. If a specific site (e.g. a cryptic splice site) cannot be inactivated due to the lack of a suitably positioned PAM, excision by segmental deletion (specified by two guides that target flanking sites) is required. A prominent clinical trial (NCT03872479) is currently in progress that uses exactly this strategy, albeit with a different Cas9 that is more prone to off-targeting than Nme2Cas9. Other loci (certain promoters/enhancers, microsatellite expansions, non-coding RNAs, etc.) are also often difficult or impossible to knock out by individual small indels and require segmental deletions. Finally, the ability to target two separate ORFs simultaneously via two individual indels adds capabilities to in vivo editing approaches. The enhanced utility of dual-sgRNA systems is therefore readily apparent to others in the field. When a hyper-accurate Cas9 such as Nme2Cas9 is employed, off-targeting potential of two guides rather than one is a readily manageable issue, as our expanded off-target analysis in this revision documents. Finally, we do not understand the referee's invocation of Cas12a (referred to by the referee by the obsolete name Cpf1), as it is impossible to induce Cas12a editing by single-AAV vector delivery.

2. In Figure 1b Dual-sgRNA design 4 produce much more full-length viral DNA than Dual-sgRNA design 1 (Judging from the band intensity and design 1 exhibited a strong band with similar size of the control—shortened single-sgRNA). However, this result doesn't match the in vivo segmental deletion efficiencies in Figure 2c where design 1 is better in both WT and Fahneo/neo model. Some explanation is needed because the choice of vector design is based entirely on the result of Figure 1b.

Response: We thank the reviewer for catching this detail. We agree that the gels of the vector genomes do not correlate with segmental deletion efficiencies. However, we note that design 1 does not significantly perform better than design 4. Nevertheless, to address this further, we have analyzed the vector genomes by nanopore sequencing to determine whether the homogeneity of genomes observed by gel can be validated. Indeed, we now show that design 4 is more homogeneous and by that metric would be expected to produce higher efficiencies of editing, again in contrast to our in vivo editing results. We conclude, perhaps not surprisingly, that additional factors beyond vector genome homogeneity can contribute to overall vector performance. To clarify this point for the readers, we have included the following statement:

“Although our nanopore sequencing data show that dual-sgRNA design 4 produced more homogeneous vector genomes than produced by design 1, our SMRT sequencing data confirms that both vectors efficiently induce segmental deletions.”

3. Using different PolIII promoter to decrease the sequence similarity is an alternative strategy to reduce vector truncation. It is better to compare different strategies.

Response: We thank the reviewer for this observation. Indeed, we examined this strategy in this system in design 2, which has one U6 promoter from mouse U6 and one U6 from human, which are different enough in sequence. The design goal for that was precisely that raised by the reviewer. While it is correct to suggest that the use of different polIII promoters can certainly improve integrity in other contexts, it appears that Design 2 has other portions that could form secondary structure, most notably the sgRNA cassette that causes the truncations. Our success with other designs obviates the need to waste time and effort with other PolIII promoters in this context.

4. In figure 1 d and e, what is the total ratio of editing efficiency, including segmental deletion inversion and indel?

(Note: Figure 1d&e in first submission are now Supplementary Fig. 1 g & h)

Response: Although we recognize the basis for the reviewer’s request, reporting the total editing efficiencies in an internally consistent manner is not so simple, based on the multiple methods needed to analyze small indels, segmental deletions and inversions. We first classified reads by their ability to map to references representing segmental deletions, inversion, or a full-length reference, and the totals were then presented as a percentage of all mapped reads (Supplementary Fig. 1g). Importantly, reads not mapped to these references were discarded and not counted, since these could be low-quality reads, sequencing/mapping errors, or reads attributed to unaccountable species. Next, reads mapping to the full-length reference only were subjected to indel analyses as described in the methods. Since reads can have sequencing errors inherent to the SMRT sequencing method and not attributed to vector design, the clustering method described in the study is the only straightforward way to group indel events that can be counted. However, the clustering method also generates a large number of singlets (reads that do not cluster), since they are poorly represented or are so low in frequency that they are considered read error. These singlets are removed from the analysis and the “high confidence” indels and unedited reads were represented as a percentages of the total counted reads (Supplementary Fig. 1h). Since the calculations to derive inversions and segmental deletions and the calculations to derive indel events have independent methods of removing reads that cannot be used to tabulate editing events, the two sets of data cannot be combined to reveal meaningful ratios of total editing events in a single representation. We feel that the separated data presented is the most accurate way of describing these events. If the reviewer

has specific suggestions to circumvent these analytical considerations, we would be happy to consider them.

5. In figure 2, why the segmental deletion ratio of mice treated with Design 1 did not increase after NTBC withdraw? In figure 2d, why the indel rate of Design 4 is largely different pre- and post-NTBC? (Increased indels only at gRNA-1 but opposite at gRNA-II). Single deletion caused by sgRNA-I is enough to disrupt Hpd gene? However, in page 8 line 18 of the third paragraph, exon 4 is excluded in 86-87% of Hpd mRNA which is probably due to gRNA-II.

Response: We thank the reviewer for this observation. Although we would indeed have been interested to see a correlation between the NGS data and the biological data, the intention of the NGS work was to validate that these indel events were carried through post-NTBC withdrawal. While our pre-NTBC analysis clearly shows that the vectors are functional in vivo, these analyses were performed in an n = 1 cohort for each vector, as collecting more samples would lead to termination of mice that would have compromised later analyses of greater importance. Overall, we believe that the biological data that we presented for this part of the study is the best indication that the intended editing was a success.

6. In Figure 3c the Nme2Cas9 expression level among design A, B, C D should be compared. An explanation is needed for the low cleavage performance of Design C.

Response: Good suggestion. We examined the mechanism of low editing performance of Design C, and it showed that intact plasmid yield using DNA miniprep for Design C was very low, for unknown reasons (see Fig. 3b).

7. Since AAV is a single stranded DNA virus, it suggests that ssDNA induces higher HDR rate than dsDNA in cells. Why two sgRNA released HDR template induced higher HDR efficiency? Moreover, two sgRNA released template could be inserted into the target site in the genome similar to HITI strategy (Nature. 2016, 540(7631):144-149). It should be analyzed.

Response: If the reviewer is referring to Fig. 3 data, we want to emphasize that this data was performed using transfection of (AAV:HDR:uncleaved and AAV:HDR:cleaved) plasmids and not AAV. Figure 3d indicates that release of dsDNA donor from the plasmid can achieve a higher HDR rate compared to donor plasmid. This is consistent with what has been reported in the literature before as free DNA ends of a released HDR template are more readily available for HDR (PMID 28219395).

We thank the reviewer for highlighting the HITI strategy. We examined our NGS data for the presence of HITI-like edited amplicons in the mice (see Supplementary Fig. 5 and Supplementary Fig. 6). It should also be noted that unsupervised detection of HITI is not a trivial task due to the lack of stable junctions at some integration sites, and to the absence of HITI analysis code to reproduce.

8. The HDR efficiency is very low in both of the disease models even for single base repair. Due to the capacity of AAV vector, HDR template could not be extend much longer for insertion of an entire ORF. Thus, what is the advantage of this system compared to base editing which achieves much high base conversion.

Response: Again, we are happy to remind the referee of some of the most basic tenets of genome editing strategies in vivo. Single-vector AAV base editing systems are poorly developed and, to our knowledge, have not been reported for successful use in vivo. Furthermore, base editing is strictly limited to certain substitutions (A-to-G, C-to-T, and, occasionally, C-to-G or -A), whereas HDR can induce all substitutions as well as indels. Base editing is also much more PAM-constrained and bystander-editing-constrained than HDR. All of these considerations

answer the referee's request for advantages over base editing. When base editing is available as an editing strategy, it is indeed advantageous over HDR in many instances, but it is simply unavailable for many desired edits, necessitating the development of alternatives. Although the referee does not invoke prime editing as another alternative with wider scope than base editing, it is similarly incompatible with single-AAV-vector delivery. HDR approaches therefore provide single-vector delivery options that neither alternative platform can match, and provide editing scope that base editing lacks.

9. In the experiment of curing the MPS-I model, the authors did not include the rAAV:ncDonor:IduaSpacer virus, which would help to exclude the NHEJ mediated read-through of the stop codon and restoring of the enzyme activity. Authors should check their NGS result to see whether there was any case like this.

Response: We agree with the suggestion that this re-examination of our NGS data would be useful and we thank the referee for raising this point. To investigate the possibility of NHEJ mediated read-through, we re-examined *Idua* amplicon sequencing from nine murine livers with high % editing and HDR. We reasoned that these samples would show the largest diversity of repair outcomes. >97% of reads from each dataset passed stringent per-base quality (≥ 30) and alignment (≥ 60) filters, yielding 45-150K reference-mapped reads per sample. 6 of 9 livers showed **no evidence of the relevant "A>G" substitution in absence of complete, desired HDR (i.e. correction of all three nucleotides via HDR donor)**. This single "A>G" substitution appeared in <350 reads in the remaining 3 livers, never exceeding 0.25% of total aligned reads. We do not believe, therefore, that the single substitution read-through would be considered a *bona fide* editing outcome as such low occurrence could be explained by sequencing or amplification errors.

10. In the MPS-1 correction assay, did the authors discovered any NHEJ mediated targeted integration? The NHEJ mediated targeted integration will also correct the point mutation but would introduce indels in the joint sites.

Response: Good suggestion -- done. See Supplementary Fig. 6.

11. MPS-1 gene therapy is a novel model, but only enzyme activity was determined. More phenotypic studies should be presented to show the efficacy.

Response: We thank the reviewer for this suggestion. It should be noted that the presence of cytoplasmic vacuoles in tissues of this MPS-I model is not obvious and cannot be observed by HE staining at 5 weeks, and (based on the literature) not even at 15 weeks of age. On the other hand, a phenotypic characteristic of lysosomal storage disease enzymes including MPS-I is upregulation of other lysosomal enzymes. We developed assays to measure the level of other lysosomal enzymes (β -D-glucuronidase, D-hexosaminidase and lysosome-associated membrane glycoprotein 1) that are commonly elevated during MPS-I prognosis. These data have been added (Fig. 5 and Supplementary Fig. 6).

Reviewers' Comments:

Reviewer #1:

Remarks to the Author:

The authors have satisfactorily addressed all major concerns

Reviewer #2:

Remarks to the Author:

The authors have revised the manuscript and addressed some of my concerns. but the novelty of the report is limited. First, the self-inactivating Cas9 system has been reported as early as 2016 (A Self-restricted CRISPR System to Reduce Off-target Effects, *Mol Ther.* 2016 Sep; 24(9): 1508–1510.) and in a paper published early this year (In vivo PCSK9 gene editing using an all-in-one self-cleavage AAV-CRISPR system, *Mol Ther Methods Clin Dev.* 2021 Mar 12; 20: 652–659.). For the use of anti-CRISPR protein to produce CRISPR/Cas9-mediated self-cleavage, a paper has been published in 2019 although an adenoviral vector was employed (Production of CRISPR/Cas9-Mediated Self-Cleaving Helper-Dependent Adenoviruses, *Mol Ther Methods Clin Dev.* 2019 Apr 16;13:432-439.). This manuscript used similar strategy in AAV production, but the principle is the same. Both SpCas9 and SaCas9 have been used for these self-cleavage CRISPR system for in vivo gene therapy. Moreover, all-in-one aav delivery of NmeCas9 has been published in 2018 (All-in-one adeno-associated virus delivery and genome editing by *Neisseria meningitidis* Cas9 in vivo, *Genome Biol.* 2018 Sep 19;19(1):137.). Although the mucopolysaccharidosis type I model is novel, the efficacy has not been substantially characterized but only observed “significant increases in IDUA activity to $3.0 \pm 0.03\%$ and $2.3 \pm 0.2\%$ of wild-type levels” and mentioned “This increase was above the therapeutic threshold required for effective reduction of GAG accumulation in the liver”. Of course enzyme activity is very important, an important reference should be cited to suggest 2-3% of WT level is sufficient for therapy. Other phenotypic reverse should be include.

RESPONSE TO CRITIQUES

We are again grateful to the reviewers for their comments. Our revision adds new information and also updates the text in ways that address the stated concerns. Specific revisions are further detailed below and denoted in red font in the manuscript text. We believe that all concerns have now been addressed and we thank the referees for helping us strengthen the study.

Reviewer #1 (Remarks to the Author):

The authors have satisfactorily addressed all major concerns.

Response: Thank you!

Reviewer #2 (Remarks to the Author):

The authors have revised the manuscript and addressed some of my concerns. but the novelty of the report is limited. First, the self-inactivating Cas9 system has been reported as early as 2016 (A Self-restricted CRISPR System to Reduce Off-target Effects, *Mol Ther.* 2016 Sep; 24(9): 1508–1510.) and in a paper published early this year (In vivo PCSK9 gene editing using an all-in-one self-cleavage AAV-CRISPR system, *Mol Ther. Methods Clin Dev.* 2021 Mar 12; 20: 652–659.).

We simply disagree, strongly, with the assertion that these earlier reports imply a lack of novelty for our work. They all fail to include many advances that our work demonstrates. When individual elements of previous reports are combined (as is always the case) in ways that have not been done before, that represents novelty, by definition. The 2016 paper did not use AAV and involved no *in vivo* editing. The 2021 paper (which appeared well after our manuscript was submitted) did not use a Cas9 with the targeting range of Nme2Cas9, did not include constructs with two guides or with guide + HDR donor, and made no mention of how self-targeting was avoided during packaging (with no indication that anti-CRISPRs were used). And this is not the full extent of our advances here – our methodologies and our readouts (including PacBio and UDiTaS analyses of editing outcomes, RNA-seq of edited tissues, Nanopore sequencing of packaged vector genomes, etc.) are also much, much more extensive and robust than these and other previous reports.

For the use of anti-CRISPR protein to produce CRISPR/Cas9- mediated self-cleavage, a paper has been published in 2019 although an adenoviral vector was employed (Production of CRISPR/Cas9-Mediated Self-Cleaving Helper-Dependent Adenoviruses, *Mol Ther Methods Clin Dev.* 2019 Apr 16;13:432-439.). This manuscript used similar strategy in AAV production, but the principle is the same.

As the author states, the earlier paper did not use the far more clinically important AAV vector system. This represents a crucial advance over the earlier work.

Both SpCas9 and SaCas9 have been used for these self-cleavage CRISPR system for *in vivo* gene therapy.

The reviewer provides no references, so it is difficult to address the claim about SpCas9. No single-vector AAV system has been reported for SpCas9, to our knowledge. As noted above, the SaCas9 work either used dual vectors rather than an all-in-one system, or it did not report how self-targeting during packaging was employed, or it did not use anti-CRISPRs for this purpose; furthermore, Nme2Cas9 has the extended targeting range afforded by a dinucleotide PAM; whereas SaCas9 does not.

Moreover, all-in-one AAV delivery of NmeCas9 has been published in 2018 (All-in-one adeno-associated virus delivery and genome editing by *Neisseria meningitidis* Cas9 in vivo, *Genome Biol.* 2018 Sep 19;19(1):137.).

This was Nme1Cas9 which has a 4-nt PAM and is very restricted in targeting range. Also, that work did not use two-vector systems nor did it include HDR nor did it use the MPS-I disease model among other differences.

Although the mucopolysaccharidosis type I model is novel, the efficacy has not been substantially characterized but only observed “significant increases in IDUA activity to $3.0 \pm 0.03\%$ and $2.3 \pm 0.2\%$ of wild-type levels” and mentioned “This increase was above the therapeutic threshold required for effective reduction of GAG accumulation in the liver”. Of course enzyme activity is very important, an important reference should be cited to suggest 2-3% of WT level is sufficient for therapy. Other phenotypic reverse should be included.

As requested, we have added a citation to an additional report [Oussoren *et al.* 2013, to go along with the Bunge *et al.* 1998 reference (ref. #91) that was already included] establishing conclusively that disease phenotypes are ameliorated by enzyme activity levels above $\sim 0.5\%$ of wild-type. Our results document statistically significant increases to 2-3% of wild-type levels, clearly demonstrating that our results exceed known therapeutic thresholds.